

# Defining and measuring microservice granularity—a literature overview

Fredy H. Vera-Rivera[1,2,3,*], Carlos Gaona[2,*] and Hernán Astudillo[4]

[1] GIA Research Group, Universidad Francisco de Paula Santander, Cúcuta, Norte de Santander, Colombia
[2] GEDI Research Group, Universidad del Valle, Santiago de Cali, Valle del Cauca, Colombia
[3] Materials and Technology Research Group, Foundation of Researchers in Science and Technology of Materials-FORISTOM, Bucaramanga, Santander, Colombia
[4] Toeska Research Team, Universidad Técnica Federico Santa María, Santiago, Chile, Chile
* These authors contributed equally to this work.

## ABSTRACT

**Background:** Microservices are an architectural approach of growing use, and the optimal granularity of a microservice directly affects the application's quality attributes and usage of computational resources. Determining microservice granularity is an open research topic.

**Methodology:** We conducted a systematic literature review to analyze literature that addresses the definition of microservice granularity. We searched in IEEE Xplore, ACM Digital Library and Scopus. The research questions were: Which approaches have been proposed to define microservice granularity and determine the microservices' size? Which metrics are used to evaluate microservice granularity? Which quality attributes are addressed when researching microservice granularity?

**Results:** We found 326 papers and selected 29 after applying inclusion and exclusion criteria. The quality attributes most often addressed are runtime properties (*e.g.*, scalability and performance), not development properties (*e.g.*, maintainability). Most proposed metrics were about the product, both static (coupling, cohesion, complexity, source code) and runtime (performance, and usage of computational resources), and a few were about the development team and process. The most used techniques for defining microservices granularity were machine learning (clustering), semantic similarity, genetic programming, and domain engineering. Most papers were concerned with migration from monoliths to microservices; and a few addressed green-field development, but none address improvement of granularity in existing microservice-based systems.

**Conclusions:** Methodologically speaking, microservice granularity research is at a Wild West stage: no standard definition, no clear development—operation trade-offs, and scarce conceptual reuse (*e.g.*, few methods seem applicable or replicable in projects other than their initial proposal). These gaps in granularity research offer clear options to investigate on continuous improvement of the development and operation of microservice-based systems.

Corresponding author
Fredy H. Vera-Rivera,
fredyhumbertovera@ufps.edu.co

## INTRODUCTION

Microservices are an architectural style that change the way applications are created, tested, implemented, and maintained. Microservices facilitate the migration of applications to the cloud infrastructure, since they allow automatic scaling, load balancing, and fault tolerance. By using microservices, a large application can be implemented as a set of small applications that can be developed, deployed, expanded, managed, and monitored independently. Agility, cost reduction and granular scalability entail some of the challenges such as the complexity of managing distributed systems (*Villamizar et al., 2015*).

The agile practices highlighting DevOps (Dev: Developers, Ops: Operations) and microservices are current trends in the development and deployment of applications in the cloud; their use by recognized technology companies and world leaders of the microservice architectural style has grown considerably. Companies such as Netflix, Amazon, eBay, PayPal, and Google, use microservices, some with more than 1,000 microservices deployed and supporting a considerable number of concurrent users. Microservice research is in a formative stage and evolution in this field is continuous. Some companies have adopted microservices to deploy their businesses. Microservices are characterized by rapid industrial adoption, and the main advances are taking place in the technology industry. However, the trend in publications and scientific advances from the academy is growing (*Soldani, Tamburri & Van Den Heuvel, 2018*).

Microservices have been presented as an implementation approach of service-oriented architecture (SOA), aiming to improve its disadvantages and problems (*Zimmermann, 2017*; *Pautasso et al., 2017*). Microservices architecture features include business orientation, polyglot programming in multiple paradigms and languages, fault-tolerant design, decentralization, and automation. Microservices are independent applications and distributed systems. SOA and microservices inherit the characteristics of distributed systems and their complexities. In this article we focus exclusively on microservices, not including SOA, web services or mobile services, which are implemented, tested, and deployed very differently than microservices.

Resilience, scalability, fast software delivery and use of fewer resources are essential characteristics of current applications. Microservices architecture came to fulfill those expectations (*Salah et al., 2016*), but many challenges still exist, such as the definition of the granularity, the complexity of managing small distributed systems, network latency and lack of reliability, fault tolerance, coherence and integration of data, distributed transaction management, communication layers, load balancing, orchestration, monitoring and security (*Pautasso et al., 2017*). The main objective of this work is to analyze the problem of defining the granularity of microservices. Microservices granularity is defined mainly, first by its size or dimensions, meaning the number of operations exposed by the microservice, along with the number of microservices that are part of the whole application, and second by its complexity and dependencies. The goal is to have low coupling, low complexity, and high cohesion between microservices. *Hassan, Bahsoon & Kazman (2020)* stated that a granularity level determines "the service size and the scope of functionality a service exposes (*Kulkarni & Dwivedi, 2008*)". Granularity adaptation

entails merging or decomposing microservices thereby moving to a finer or more coarse-grained granularity level (*Hassan, Bahsoon & Kazman, 2020*). *Homay et al. (2020)* stated that "the problem in finding service granularity is to identify a correct boundary (size) for each service in the system. In other words, each service in the system needs to have a concrete purpose, as decoupled as possible, and add value to the system. A service has a correct or good granularity if it maximizes system modularity while minimizing the complexity. Modularity in the sense of flexibility, scalability, maintainability, and traceability, whereas complexity in terms of dependency, communication, and data processing" (*Homay et al., 2020*).

The quality of a microservices-based system is influenced by the granularity of its microservices, because their size and number directly affect the system's quality attributes. The optimal size or granularity of a microservice directly affects application performance, maintainability, storage (transactions and distributed queries), and usage and consumption of computational resources (mainly in the cloud, the usual platform to deploy and execute microservices). Although the size of microservice or optimal granularity is a discussion topic, few patterns, methods, or models exist to determine how small a microservice should be, as others have already pointed out:

*Soldani, Tamburri & Van Den Heuvel (2018)* noticed the difficulty of identifying the business capacities and delimited contexts that can be assigned to each microservice (*Soldani, Tamburri & Van Den Heuvel, 2018*). *Bogner, Wagner & Zimmermann (2017a)* claimed that "the appropriate microservice granularity is still one of the most discussed properties (How small is small enough?), as shown in the difficulty of defining acceptable value ranges for source code metrics" (*Bogner, Wagner & Zimmermann, 2017a*). *Zimmermann (2017)* indicated that professionals request more concrete guidance than the frequent advice to "define a limited context for each domain concept that will be exposed as a service" (*Zimmermann, 2017*). *Jamshidi et al. (2018)* affirmed that the real challenge is finding the right modules, with the correct size, the correct assignment of responsibilities, and well-designed interfaces, and besides, no agreement on the correct size of microservices exist (*Jamshidi et al., 2018*).The aim of this article is to identify the main approaches in the literature that define microservice granularity or that use it in the process of designing microservice-based systems, either from scratch or migrated from monoliths. A systematic literature review was carried out on key scientific computing literature databases (IEEE Xplore, ACM Digital Library, and Scopus); we formulated three research questions, we defined inclusion and exclusion criteria, and current research trends to identify relevant works, challenges and gaps for future research were identified. Research papers that address the problem of microservice granularity are detailed; the research questions are (RQ1) which approaches have been proposed to define microservice granularity? (RQ2) which metrics are used to evaluate microservice granularity? and (RQ3) which quality attributes are addressed when researching microservice granularity? Very few previous works have reviewed the definition of microservices granularity; we did not find any review that details the techniques, methods or methodologies used to define granularity, none describe the metrics used to evaluate it, and few addresses the quality attributes considered to define it.

Contributions of this work are as follow: (1) we identified and classified research papers that address the problem of microservice granularity, therefore we defined the state of the art; (2) we identified and defined the metrics currently used to assess the granularity of microservices-based systems; (3) we identified the quality attributes that researchers studied to define microservice granularity; (4) we identified the case studies used to validate the methods, which can serve as a dataset for future evaluations of methods or techniques to define granularity.

The remainder of this article is organized as follows: 'Related Work' defines previous related works; 'Survey Methodology' presents the survey design; 'Results' organizes the results; 'Discussion' discusses the trends and research gaps; and 'Conclusions' summarizes and concludes.

## RELATED WORK

A small number of literature reviews have been published on microservice architecture research, some of these papers include analysis of application modeling and architecture; design and development patterns; industrial adoption; state of practice; grey literature review; and analysis and interviews with industry leaders, software architects, and application developers of microservices-based applications; whereas two papers focused on microservice granularity specifically (*Hassan, Bahsoon & Kazman, 2020*; *Schmidt & Thiry, 2020*). The literature reviews are described in chronological order below.

*Di Francesco (2017)* and *Di Francesco, Lago & Malavolta (2019)* focused on determining the publication trends of research studies on architecture with microservices and on the potential for industrial adoption of existing research. They point out that few studies have focused on design patterns and architectural languages for microservices, and research gaps exist in areas related to quality attributes (*Di Francesco, 2017*; *Di Francesco, Lago & Malavolta, 2019*).

*Zimmermann (2017)* extracted the principles of microservices from the literature and makes a comparison with SOA and highlights the critical points in the research on microservices, as a result of the review and discussions with industry opinion leaders, developers and members of the service-oriented community. He raises five research issues: (1) service interface design (contracting and versioning), (2) microservice assembly and hosting, (3) microservice integration and discovery, (4) service dependency management, and (5) service and end client application testing (*Zimmermann, 2017*).

*Jamshidi et al. (2018)* presented a technological and architectural perspective on the evolution of microservices. Their editorial introduction also set out future research challenges: (1) service modularization and refactoring; (2) service granularity; (3) front-end integration; (4) resource monitoring and management; (5) failure, recovery and self-repair; and (6) organizational culture and coordination (*Jamshidi et al., 2018*).

*Soldani, Tamburri & Van Den Heuvel (2018)* systematically analyzed the grey industrial literature on microservices to define the state of practice, identified the technical and operational problems and benefits of the architectural style based on microservices at an industrial level. When designing a microservice-based application, key issues involve determining the right granularity of its microservices and the design of its security policies.

During development time, managing distributed storage and application testing, is challenging. Another pain was usage of network and computing resources during operation (*Soldani, Tamburri & Van Den Heuvel, 2018*).

*Ghofrani & Lübke (2018)* focused on identify challenges and gaps in the design and development of microservices, they describe the main reasons leveraging and preventing the usage of systematic approaches in microservice architectures; and the suggestions or solutions that improve aspects of the microservices architecture. *Ghofrani & Lübke (2018)* provided an updated map of the state of practice in microservice architecture and its complexities for future research. According to the results of their survey, optimization in security, response time, and performance had higher priorities than resilience, reliability, fault tolerance, and memory usage are research gaps.

*Osses, Márquez & Astudillo (2018)* summarized the 44 architectural patterns of microservices, and proposed a microservice architectural pattern taxonomy: front-end, back-end, orchestration, migration, internet of things, and DevOps. There was no specific pattern to define the adequate microservice granularity, and they just proposed designing the application as a set of modules, each one an independent business function with its data, developed by a separate team and deployed in a separate process (*Osses, Márquez & Astudillo, 2018*).

*Hamzehloui, Sahibuddin & Salah (2019)* aimed to identify the common trends and direction of research in microservices. They stated that infrastructure-related issues were more common than software-related issues, and the cloud was the most common platform for running microservices. At the infrastructure level, automation and monitoring require more research, as do software development and design in microservices; safety, maintenance, and costs were three other areas that have been studied relatively less compared to other topics (*Hamzehloui, Sahibuddin & Salah, 2019*).

*Vera-Rivera, Gaona Cuevas & Astudillo (2019)* identified the challenges and research trends present in the phases of the development process and in the management of quality attributes of microservice-based applications (*Vera-Rivera, Gaona Cuevas & Astudillo, 2019*). This article was more general, it did not emphasize in granularity.

*Hassan, Bahsoon & Kazman (2020)* carried out a systematic mapping study to provide a better understanding of the transition to microservices; they consolidated various views (industrial, research/academic) of the principles, methods, and techniques commonly adopted to assist in the transition to microservices. They identified gaps in the state of the art and the practice related to reasoning about microservice granularity. In particular, they identified possible research topics concerning (1) systematic architecture-oriented modeling support for microservice granularity, (2) a dynamic architectural assessment approach for reasoning about the cost and benefit of granularity adaptation, and (3) effective decision support for informing reasoning about microservice granularity at runtime (*Hassan, Bahsoon & Kazman, 2020*). They focused on understanding the transition to microservices and the microservice granularity problem (a direct antecedent of this study). They considered quality attributes but not metrics. Their sources were gray literature (blog articles, presentations, and videos, as means of reporting first-hand industrial experiences) and research papers, whereas our study put emphasis only on white

literature (articles published in journals and scientific events). Our work was more specific and detailed evaluating the methods and techniques to define granularity, whereas their work detailed those methods and techniques in general. Our work are complementary to their work and take a deeper look at the definition of granularity.

*Schmidt & Thiry (2020)* carried out a systematic literature review, they found proposals for identification, decomposition, partitioning or breaking down the application domain to reach an adequate granularity for microservices. Moreover, the research aims to highlight the usage of Model-Driven Engineering (MDE) or Domain-Driven Design (DDD) approaches (*Schmidt & Thiry, 2020*). They emphasized on DDD and MDE; whether the selected studies cover DDD or apply MDE; and which elements, principles, practices, and patterns authors applied; they did not include metrics and quality attributes. Therefore, our work is complementary to their work.

Most previous literature reviews do not emphasize granularity, they concern general topics of microservice architecture. To our knowledge, this is the first study focuses specifically on classified and detailed research papers of microservice granularity, including quality attributes that motivate working on it, methods/techniques to improve it, and metrics to measure it.

## SURVEY METHODOLOGY

A systematic literature review was carried out following the approach introduced by Kitchenham define systematic literature reviews as "a form of secondary study that uses a well-defined methodology to identify, analyze and interpret all available evidence related to a specific research question in a way that is unbiased and (to a degree) repeatable" (*Kitchenham, 2004*).

### Planning the systematic literature review

The objectives of this systematic literature review are defined as follows: first, to identify the proposals that address the microservice granularity problem; second, to identify the metrics that have been used to evaluate microservice granularity; and third, to analyze the quality attributes addressed in those works to evaluate microservice granularity. Few studies or reviews specifically address the problem of microservice granularity, and very few identify the metrics along with the quality attributes addressed to assess microservice granularity.

A review protocol specifies the methods that will be used to undertake a specific systematic review. We selected research papers through two queries strings used in the IEEE Xplore, ACM Digital Library, and Scopus; then the papers were screening and reviewed, we applied inclusion and exclusion criteria, next we tabulated the papers, the contribution of each selected paper was detailed, we classified the papers, the metrics were described, and the quality attributes were specified. The protocol components are listed below:

1. Define research questions.
2. Search Strategy.

3. Data extraction strategy

4. Study selection criteria and procedures.

5. Synthesis of the extracting data.

**1. Define the research questions.** The research questions (RQs) covered in this systematic literature review were:

RQ1: Which approaches have been proposed to define microservice granularity?

RQ2: Which metrics are used to evaluate microservice granularity?

RQ3: Which quality attributes are addressed when researching microservice granularity?

The definition of the correct size and the functionalities that each microservice must contain, affects the quality attributes of the system, and affects testing, deployment, and maintenance; therefore, identify the metrics that are being used to evaluate the granularity is very important, describe them and use them as references to propose other methods, techniques or methodologies that allow defining the granularity of the microservices in a more objective way. Identifying the quality attributes studied and how they were evaluated is an important reference for future work to define the granularity of the microservices that are part of an application.

**2. Search strategy:** Two query strings were defined, which included alternative spellings for "microservice" and any of the following words: granularity, size, dimensioning, or decomposition. Additionally, to make the search terms that correspond to the research questions more precise, another search string was included. In addition to granularity, it contained the words "method", "technique", and "methodology". Therefore, the search strings were as follow:

**Query string 1 (QS1):** ("micro service" OR microservice) AND (granularity OR size OR dimensioning OR decomposition).

According to *Hassan, Bahsoon & Kazman (2020)*, it is established that the granularity of microservices is related to the size and dimension of the microservice, so these terms are included in the search string. Additionally, the decomposition of monolithic applications to microservices is an important research topic, so we include this word.

**Query string 2 (QS2):** ("micro service" OR microservice) AND "granularity" AND ("method" OR "technique" OR "methodology"), targeting only research papers.

The main objective of the work was to identify the methods, techniques or methodologies used to determine the microservices granularity.

QS1 and QS2 addresses all research question; for each of the proposals selected in each QS, the metrics used, and the quality attributes addressed were identified. The query strings were used in IEEE Xplore[1], ACM Digital Library[2] and Scopus[3], searching for papers' titles, abstracts and keywords. The search in these databases, yield 969 results for QS1 and 146 results for QS2. The search was performed in July 2020.

**3. Data extraction strategy.** First, papers were tabulated; second, duplicated papers were removed; third, title, abstract, and conclusions of all papers were reviewed and analyzed. Each coauthor of this report carried out this process.

---

[1] IEEE Xplore: https://ieeexplore.ieee.org/

[2] ACM Digital Library: https://dl.acm.org/

[3] Scopus: https://www.scopus.com/search/form.uri?display=basic

**4. Study selection criteria and procedures.** We selected primary research papers that make a specific proposal (methodology, model, technique, or method) about microservices granularity, including migrations from monolith to microservices and decompositions of systems in microservices. After obtaining the relevant studies, the inclusion and exclusion criteria were applied (see Table 1). We excluded any paper about monolith migrations that were not directly related to the definition of microservice granularity. We also excluded papers that proposed methods, techniques, or models for SOA, web services or mobile services.

**5. Synthesis of the extracting data.** Each of the selected papers was evaluated in full-text form, taking detailed note of the work and the contributions made.

## Conducting the systematic literature review

The review process was carried out as follow:

1. We download the full-text paper.
2. Each co-author read and review the paper.
3. Each co-author uses the classification criteria on the paper, using the table presented in the Appendix A; this was carried out by each co-author independently.
4. We discussed and analyzed the results obtained by each author, resolving doubts and contradictions and the results are presented in Appendix A.

The results of applying the research protocol is presented in Fig. 1. To analyze the works presenting definitions of the granularity of microservices, classification criteria were defined. These criteria were based on the classification performed by (Wieringa et al., 2006), and have been widely used in previous systematic literature reviews: (Di Francesco, Lago & Malavolta, 2019; Hamzehloui, Sahibuddin & Salah, 2019; Hassan, Bahsoon & Kazman, 2020; Vural, Koyuncu & Guney, 2017). To answer the research questions, we added the classification criteria in each paper: metrics, stage of the development process, technique used, and quality attributes studied or analyzed; namely:

- **Metrics used:** Which metrics are used to define the granularity of microservices?
- **Development process phases:** Phases of the development process on which the work focuses.
- **Research strategies:** Includes solution proposal, validation research, experience paper, opinion paper, philosophical paper, and evaluation research.
- **Approach:** Structural or behavioral aspects proposed in the papers to define the granularity of microservices (Hassan, Bahsoon & Kazman, 2020).
- **Quality attribute studied:** The Quality attributes considered in the proposal, such as performance, availability, reliability, scalability, maintainability, security, and complexity.
- **Research contribution:** Type of contribution made in the article; namely, method, application, problem formulation, reference architecture, middleware, architectural language, design pattern, evaluation, or comparison.

- **Experimentation type:** Type of experimentation used to validate the proposal; namely experiment, case study, theoretical, or other.
- **Technique used:** This criterion describes the technique, method or model used to define the granularity of the microservices.
- **Input data:** Type of input data used to identify the microservices (*i.e.*, uses cases, logs, source code, execution traces, among others)
- **Type of case study:** This criterion determines if the case study is a toy example (hypothetical case) or a real-life case study. We identified the case study.
- **Automatization level:** This criterion determines the level of automation of the proposed technique, if it is manual, automatic, or semi-automatic.

Finally, results were presented in four sections: first, the classification of the selected papers; second, the main contributions and research gaps in sizing and definition of microservice granularity were detailed; third, metrics were described an ordered by year and type; and fourth, quality attributes were detailed, and results were discussed, leading to conclusions presented in this article.

## RESULTS

The search process took place in July 2020. The search in the databases of scientific publications when applying the search strings (QS1 and QS2) related to the granularity of the microservices yield 969 and 146 works respectively (see Table 2).

After applying the inclusion and exclusion criteria, 29 papers were selected that address the definition of the granularity of microservices. (see Table 3). The summarized results of this systematic literature review are synthesized in Fig. 2.

For RQ1, we identified the papers that propose a method, model, or methodology to define the microservice granularity; metrics are fundamental because they allow one to measure, monitor, and evaluate any aspect of a microservice, thus defining or determining the appropriate granularity of a microservice. For RQ2, we identified metrics used to evaluate microservice granularity and their decomposition. Figure 2 shows the type and number of metrics and whether it was applied to microservice, system, development process, or development team. These metrics are detailed in 'RQ2: Metrics to Evaluate the Microservice Granularity'. Finally, for RQ3 we synthesized the works that address quality attributes to evaluate microservices granularity.

### Classification of the selected papers

Appendix A shows the tabulated data and the results of the evaluation of classification criteria. Most papers were published in conference (86%), and only four (14%) were published in journals. All selected papers were published between 2016 and the beginning of 2020 (two in 2016, seven in 2017, six in 2018, 12 in 2019, and two in 2020).

The development process phases addressed by each proposal are shown in Fig. 3. Several papers emphasize more than one phase, (*e.g.*, P10 focuses on development and deployment, as befits a method for migration from monolith to microservices).

**Table 1 Inclusion and exclusion criteria.**

| Inclusion criteria | Description |
| --- | --- |
| Primary research papers that make a specific proposal about the size, granularity, or decomposition of applications to microservices. | This criterion focuses on identifying primary research papers that propose or define the size or granularity of microservices, also we include migrations from monolith to microservices that carry out a proposal to decompose the monolithic application to microservices. |
| Papers that propose a methodology, model, technique, or method to define granularity, size, or dimension of microservices. | The objective of the review is to identify the models, methods, methodologies, or techniques used to define the microservice granularity. |
| Migrations that include a methodology, model, technique, or method to define granularity, size, or dimension of microservices. | We include migrations from monolithic applications to microservices that reason about the definition of microservice granularity, those migrations that focus on other aspects are not included. |
| Papers published in journals and conference proceedings in the field of software architecture, software engineering and computer science. | We focus on research papers published in international journals and conferences only in software architecture, software engineering, and computer science. We include only peer-reviewed papers. We did not include gray literature. |

| Exclusion criteria | Description |
| --- | --- |
| Tutorial, example, experience, and opinion articles. | We do not include tutorials, examples, experiences, and opinion articles, because they do not correspond to primary research papers, or they do not carry out a new contribution in the definition of microservices granularity. |
| Survey and literature review. | We exclude survey papers, and literature reviews because they are secondary research papers that list the contributions of other authors. |
| Use of microservices in other areas. | The use of microservices architecture in other areas is evident and fundamental, for this review they were excluded because they do not directly address the problem of defining the microservice granularity. |
| Papers that do not include a methodology, model, technique, or method to define granularity, size, or dimension of microservices. | Articles related to the microservice architecture, which do not make a specific proposal on the definition of microservices granularity are excluded. |
| Papers which propose a specific method, technique or model for SOA, web services or mobile services. | The fundamentals of SOA, web services, and mobile services are different from the fundamentals of microservices architecture, so specific proposals in these topics are not included. |
| Literature only in the form of abstracts, blogs, or presentations. | We used full-text articles, excluding those that are only available in abstract, blog, or presentation form (not peer-reviewed). |
| Articles not written in English or Spanish. | We only include papers written in English or Spanish; other languages are excluded. |

Most of the proposed methods focus on the design (79%) and development (38%) phase, with only one addressing testing (3%). Migrations from monolithic architectures to microservices are very common and important 19 of 29 papers (66%). The papers that do not address migration do focus on identifying microservices in the design phase; therefore, defining the size and granularity of microservices from the design phase on is key, because it has implications for development, testing and deployment.

Further, most papers (79%) focus on the design phase; implicitly or explicitly, they suggest that defining the "right" microservices granularity from the design phase on is fundamental. However, some authors affirm that reasoning about microservices size and performance is not possible at design time; indeed, (*Hassan, Ali & Bahsoon, 2017*) affirm that the expected behavior of the system cannot be fully captured at design time.

On the research strategy (see Fig. 4), validation research and solution proposals account for almost all (14 and 11, respectively); proposals that have been tested and validated in

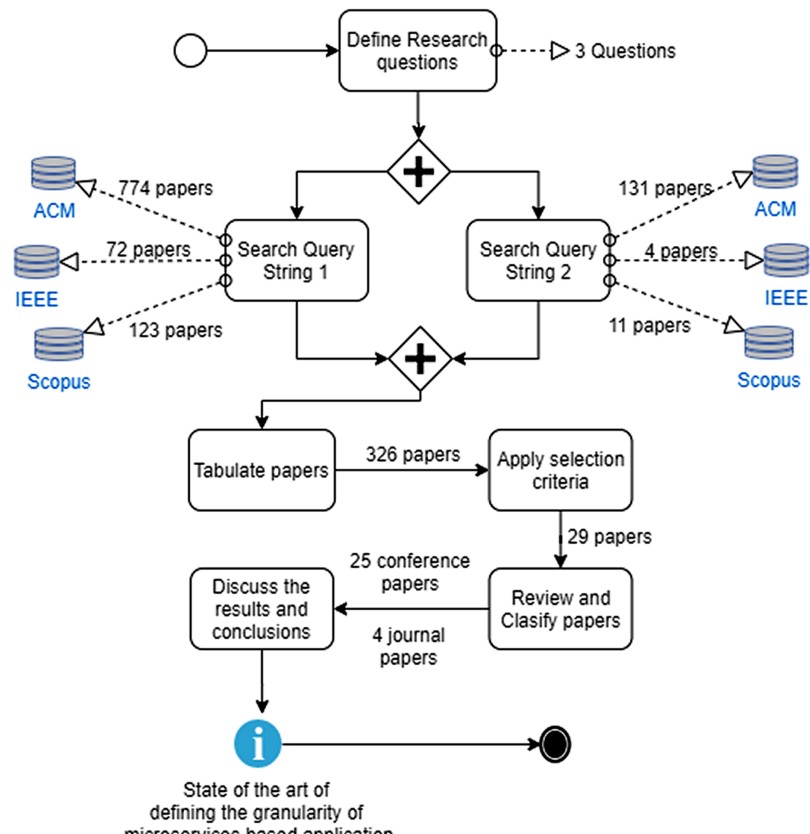

**Figure 1 Systematic literature review method.**

**Table 2 Number of publications.**

| Query String | Mont/Year | IEEE | ACM | Scopus | Total* |
|---|---|---|---|---|---|
| QS1 | July/2020 | 72 | 774 | 123 | 969 |
| QS2 | July/2020 | 4 | 131 | 11 | 146 |

**Note:**
* Including duplicated papers.

practice are very few, namely P5 (a reference architecture) and P16 (a method for candidates microservice identification from monolithic systems).

On the type of contribution (see Fig. 5), the vast majority (17 papers) proposed methods (59%), some proposed methodologies (24%), few proposed reference architectures (7%) and problem formulation (7%), Only one propose an evaluation or comparison (3%).

On the validation approach (see Fig. 6), most papers (69%) used case studies for validation and evaluation, other papers use experiments (37%), and most of also used case studies.

More than half of studies (13 of 29) validated their proposals using realistic (but not real—hypothetical) case studies, and the remaining almost-half (14 of 29) used real-life case studies, real-life case studies achieve better validation than hypothetical case studies.

**Table 3 Selected papers, related works on the definition of granularity of microservices.**

| ID. | Paper | Year | Type |
|---|---|---|---|
| P1 | Microservices and Their Design Trade-offs: A self-adaptive Roadmap (*Hassan & Bahsoon, 2016*). | 2016 | Conference paper |
| P2 | Microservice Architectures for Scalability, Agility and Reliability in E-Commerce (*Hasselbring & Steinacker, 2017*). | 2017 | Conference paper |
| P3 | From Monolith to Microservices: Lessons Learned on an Industrial Migration to a Web Oriented Architecture (*Gouigoux & Tamzalit, 2017*). | 2017 | Conference paper |
| P4 | Microservices: Granularity *vs*. Performance (*Shadija, Rezai & Hill, 2017*). | 2017 | Conference paper |
| P5 | Microservice Ambients: An Architectural Meta-Modelling Approach for Microservice Granularity (*Hassan, Ali & Bahsoon, 2017*). | 2017 | Conference paper |
| P6 | Microservices Identification Through Interface Analysis (*Baresi, Garriga & De Renzis, 2017*). | 2017 | Conference paper |
| P7 | Partitioning Microservices: A domain engineering approach (*Josélyne et al., 2018*). | 2018 | Conference paper |
| P8 | A Case Study on Measuring the Size of Microservices (*Vural, Koyuncu & Misra, 2018*) | 2018 | Conference paper |
| P9 | Identifying Microservices Using Functional Decomposition (*Tyszberowicz et al., 2018*). | 2018 | Conference paper |
| P10 | Unsupervised Learning Approach for Web Application Auto-decomposition into Microservices (*Abdullah, Iqbal & Erradi, 2019*). | 2019 | Journal paper |
| P11 | Requirements Reconciliation for Scalable and Secure Microservice (De)composition (*Ahmadvand & Ibrahim, 2016*). | 2016 | Conference paper |
| P12 | Function-Splitting Heuristics for Discovery of Microservices in Enterprise Systems (*De Alwis et al., 2018*). | 2018 | Conference paper |
| P13 | Extraction of Microservices from Monolithic Software Architectures (*Mazlami, Cito & Leitner, 2017*). | 2017 | Conference paper |
| P14 | From Monolith to Microservices: A Dataflow-Driven Approach (*Chen, Li & Li, 2017*). A Dataflow-driven Approach to Identifying Microservices from Monolithic Applications (*Li et al., 2019*). | 2017 2019 | Conference paper Journal paper |
| P15 | From Monolithic Systems to Microservices: A Decomposition Framework Based on Process Mining (*Taibi & Syst, 2019*). | 2019 | Conference paper |
| P16 | Service Candidate Identification from Monolithic Systems Based on Execution Traces (*Jin et al., 2019*). | 2019 | Journal paper |
| P17 | The ENTICE Approach to Decompose Monolithic Services into Microservices (*Kecskemeti, Marosi & Kertesz, 2016*). Towards a Methodology to Form Microservices from Monolithic Ones (*Kecskemeti, Kertesz & Marosi, 2017*). | 2016 2017 | Conference paper |
| P18 | Refactoring Orchestrated Web Services into Microservices Using Decomposition Pattern (*Tusjunt & Vatanawood, 2018*). | 2018 | Conference paper |
| P19 | A logical architecture design method for microservices architectures (*Santos et al., 2019*). | 2019 | Conference paper |
| P20 | A New Decomposition Method for Designing Microservices (*Al-Debagy & Martinek, 2019*). | 2019 | Journal paper |
| P21 | Business Object Centric Microservices Patterns (*De Alwis et al., 2019*). | 2019 | Conference paper |
| P22 | From a Monolith to a Microservices Architecture: An Approach Based on Transactional Contexts (*Nunes, Santos & Rito Silva, 2019*). | 2019 | Conference paper |
| P23 | Granularity Cost Analysis for Function Block as a Service (*Homay et al., 2019*). | 2019 | Conference paper |
| P24 | MicroValid: A Validation Framework for Automatically Decomposed Microservices (*Cojocaru, Uta & Oprescu, 2019*). | 2019 | Conference paper |
| P25 | Migration of Software Components to Microservices: Matching and Synthesis (*Christoforou, Odysseos & Andreou, 2019*). | 2019 | Conference paper |
| P26 | Microservice Decomposition *via* Static and Dynamic Analysis of the Monolith (*Krause et al., 2020*). | 2020 | Conference paper |
| P27 | Towards Automated Microservices Extraction Using Multi-objective Evolutionary Search (*Saidani et al., 2019*). | 2019 | Conference paper |
| P28 | Extracting Microservices' Candidates from Monolithic Applications: Interface Analysis and Evaluation Metrics Approach (*Al-Debagy & Martinek, 2020*). | 2020 | Conference paper |
| P29 | Migrating Web Applications from Monolithic Structure to Microservices Architecture (*Ren et al., 2018*). | 2018 | Conference paper |

Even better, some studies (8) used actual open-source projects. The case studies found in the reviewed articles are summarized in Table 4; they are valuable resources to validate future research and to compare new methods with those identified in this review.

In any case, other microservice-based datasets have been found to be beyond the reach of this study; for example, (*Rahman, Panichella & Taibi, 2019*) shared a dataset composed of 20 open-source projects using specific microservice architecture patterns; and (*Marquez & Astudillo, 2018*) shared a dataset of open source microservice-based projects when investigating actual use of architectural patterns.

The most used case studies to validate the proposals were Kanban boards (P6, P20, P28) and Money transfer (P6, P20, P28); they were used by three papers, followed by JPetsStore (P16, P29) and Cargo tracking (P6, P24) which was used by two papers.

## RQ1: Approaches to define microservices granularity

The granularity of microservices involves defining their size and the number that will be part of the application. From the proposal of *Newman (2015)*, microservices follow the principle of simple responsibility that says "Gather things that change for the same reason and separate things that change for different reasons". The size, dimension, or granularity of microservices have traditionally been defined as follows:

1. Trial and error, depending on the experience of the architect or developer.
2. According to the number of lines of code.
3. By implementation units.
4. By business capabilities.
5. By capabilities of the development team or teams.
6. Using domain-driven design.
7. Number of methods or exposed interfaces.

*Richardson (2020)* proposed four decomposition patterns, which allow for the decomposition of an application into services: (1) Decompose by business capability: define services corresponding to business capabilities; (2) decompose by subdomain: define services corresponding to DDD subdomains; (3) self-contained service: design services to handle synchronous requests without waiting for other services to respond. (4) service per team: each service is owned by a team, which has sole responsibility for making changes, and ideally each team has only one service (Richardson & microservices.io).

*Zimmermann et al. (2019)* proposed a microservice API patter (MAP) for API design and evolution. The patterns are divided in five categories: (1) foundation, (2) responsibility, (3) structure, (4) quality, and (5) evolution. These patterns are an important reference for developing microservice-based applications. There is no specific pattern that helps to determine the number of microservices and their size, that is, the number of operations it must contain (*Zimmermann et al., 2019*).

The size of the microservice or optimal granularity is one of the most discussed properties and there are few patterns, methods, or models to determine how small a microservice should be. In this respect, some authors have addressed this problem and proposed the solutions summarized in Table 5.

The proposed techniques were classified into manual, semi-automatic, or automatic techniques; manual techniques are methods, procedures, or methodologies performed by

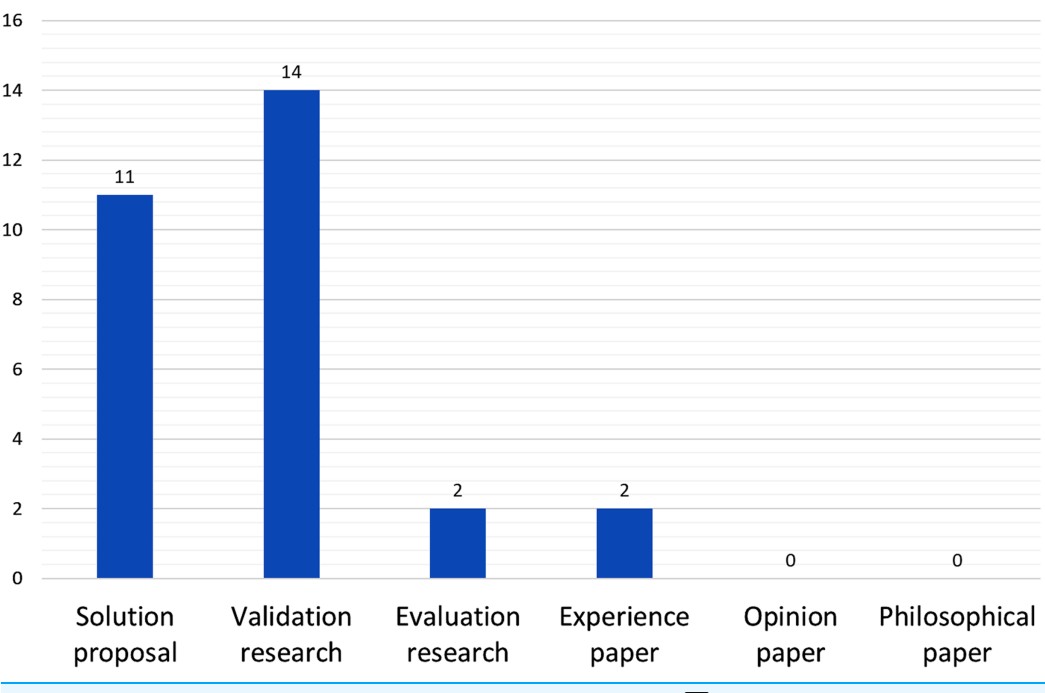

**Figure 4 Number of papers by research strategy.**

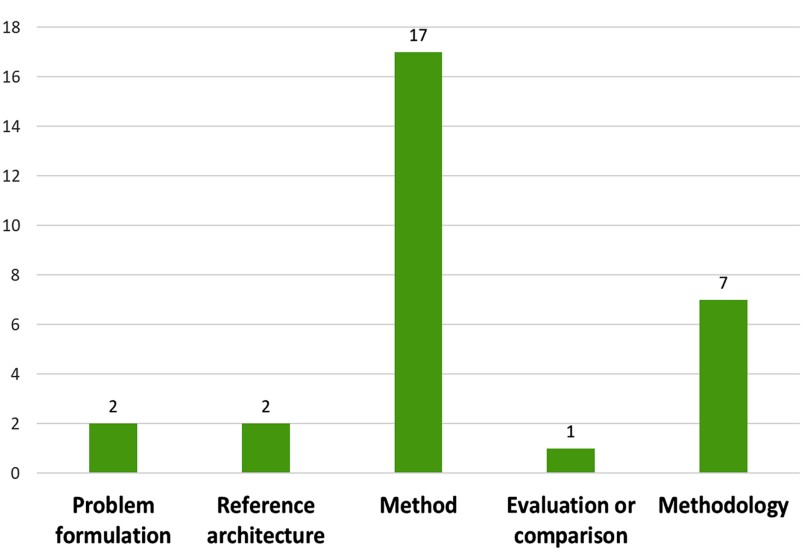

**Figure 5 Number of papers by contribution type.**

the architect or developer decomposing systems following a few steps. Automatic techniques use some type of algorithm to generate decomposition, and the system generates the decomposition. Semi-automatic combine one part made manually and with another made automatically. Most papers proposed manual procedures to identify the microservice granularity (15 papers); some proposals were automatic (eight papers) and

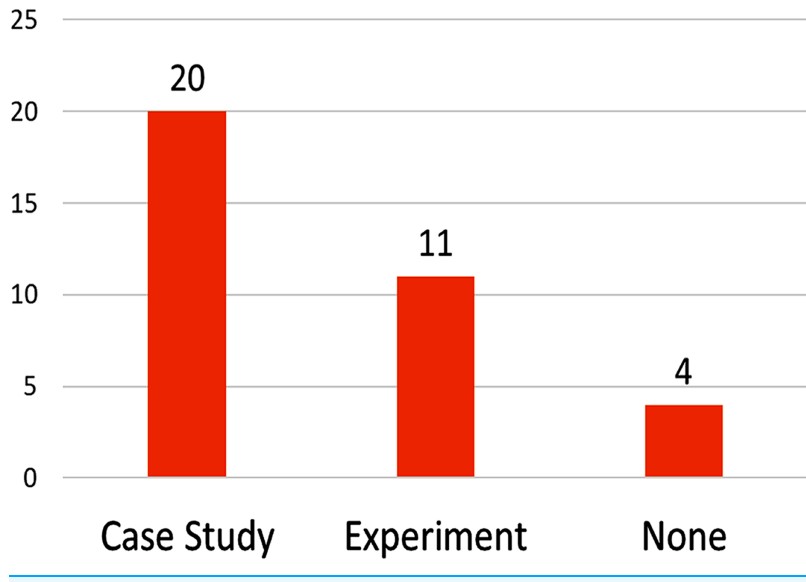

**Figure 6** **Number of papers by validation approach.**

few proposals paper were semi-automatic (six papers). The most used case studies to validate the proposals were Kanban boards and Money Transfer (P6, P20, P28).

The papers from 2017 and 2018 are mostly manual methods or methodologies that detail the way to decompose or determine microservices, using DDD, domain engineering, or a specific methodology. Later, the papers from 2019, and 2020 propose semi-automatic, and automatic methods that use intelligent algorithms and machine learning mostly focused on migrations from monolith to microservices. We can observe a chronological evolution in the proposals, the type of techniques used to define the granularity of the microservices that are part of an application are presented in Fig. 7, semantic similarity, machine learning, and genetic programing were the most important techniques.

## RQ2: Metrics to evaluate the microservice granularity

Software metrics allow us to measure and monitor different aspects and characteristics of a software process and product, and there are metrics at the level of design, implementation, testing, maintenance, and deployment. These metrics allow understanding, controling, and improving of what happens during the development and maintenance of the software, to take corrective and preventive actions.

In the methods and models identified, most of them (59%) used some metrics to determine microservices granularity. We would have expected greater importance for metrics in automatic methods to validate granularity in microservice-based applications and evaluate decompositions yield by methods.

We identify metrics for coupling, cohesion, granularity, complexity, performance, use of computational resources, development team, source code, and so on (see Table 6). We classified them into four groups: about development team, about microservices development process, about the system, and about each microservice. Most identified metrics (40) focused on a microservice, and only two address the microservice

**Table 4 Types of case study used by selected papers.**

| Paper | Case Study | Type | URL |
|---|---|---|---|
| P2 | An e-commerce web application called otto. | Real-life | http://www.otto.de |
| P3 | MGDIS. | Real-life | http://www.mgdis.fr |
| P4 | University admissions systems. Central Admission Authority UK. | Hypothetical | No report |
| P5 | Online movie streaming subscription-based system. | Hypothetical | No report |
| P6 | Cargo tracking, money transfer, and kanban boards. | Hypothetical | No report |
| P6 | Real-world OpenAPI specification, from Apis.guru | Real-life | https://apis.guru/openapi-directory/ |
| P7 | Weather case study. | Hypothetical | No report |
| P8 | E-shop | Hypothetical | No report |
| P9 | The Common Component Modeling Example–CoCoME *Herold et al (2008)*. | Hypothetical | No report |
| P10 | Open source benchmark web application ACME Air. | Hypothetical | https://github.com/blueperf, https://github.com/blueperf/acmeair-driver |
| P11 | Movie/TV show streaming system. | Hypothetical | No report |
| P12 | Open-source projects: SUGAR and Church CRM. | Real-life | https://www.sugarcrm.com/. http://churchcrm.io/ |
| P13 | Open source repositories using the Git VCS. (21 projects) | Real-life | No report |
| P14 | Case study 1: movies and comments. Case study 2: cinema, movies, plan screening, online box, office list. | Hypothetical | No report |
| P15 | Bookkeeping for Italian tax accountants. | Real-life | No report |
| P16 | JPetsStore, a web application. | Real-life | https://github.com/mybatis/jpetstore-6 |
| P16 | SpringBlog. Solo. jforum2. Agilefant. Xwiki. | Real-life | https://github.com/bvn13/SpringBlog https://github.com/b3log/solo https://sourceforge.net/projects/jforum2 https://roller.apache.org https://www.agilefant.com http://platform.xwiki.org |
| P18 | Online shopping workflow. | Hypothetical | No report |
| P19 | The Integrated Management Platform 4.0 (IMP_4.0), ERP System. | Real-life | No report |
| P20 | Kanban boards. Money transfer. | Hypothetical | No report |
| P20 | AWS. PayPal. | Real-life | No report |
| P22 | LdoD. Blended workflow. | Real-life | https://ldod.uc.pt https://github.com/socialsoftware/blended-workflow. |
| P24 | Cargo tracking. Booking system. Trading system. Movie crawler system. Ticket price comparator. | Hypothetical | No report |
| P26 | Legacy lottery application in\|FOCUS. | Real-life | No report |
| P27 | JpetStore, SpringBlog | Hypothetical | https://github.com/mybatis/jpetstore-6. 2 https://github.com/Raysmond/SpringBlog. |
| P28 | Kanban boards. Money transfer. | Hypothetical | No report |
| P29 | DayTrader, JPetsStore, TPC-W. RUBiS. 12 applications migrated. | Real-life | No report |

development processes. There is a research gap for metrics to evaluate the full development process of microservice-based applications and their impact on the granularity of microservices.

The most used metrics are related to coupling (14 proposed metrics), followed by performance and cohesion (13 metrics), next, computational resources metrics (eight metrics), and complexity and source code (seven metrics); (see Fig. 8). Nine papers used coupling metrics (P11, P12, P13, P14, P15, P16, P21, P22, P24), and seven papers used cohesion metrics (P12, P14, P16, P21, P24, P27, P28), whereas performance metrics was used by five papers (P4, P10, P12, P21, P23); Complexity metrics were considered by two papers (P8, P22), although they are fundamental characteristic of microservices.

More proposals that include more complexity metrics are required, as well as metrics related to the microservice development process. The other metrics were used by only one paper each. We found that 11 papers used coupling or cohesion metrics, and 5 papers used both. Only one (P24) used coupling, cohesion, and complexity metrics.

The size and number of microservices that compose an application directly affects its maintainability. Automation of tests, continuous integration and deployment are essential especially when microservices and many distributed systems must be managed independently by each microservice.

*Bogner, Wagner & Zimmermann (2017b)* performed a literature review to measure the maintainability of software and identified metrics in four dominant design properties: size, complexity, coupling, and cohesion. For service-based systems, they also analyzed its application to systems based on microservices and presented a maintainability model for services (MM4S), consisting of service properties related to automatically collectible service metrics (*Bogner, Wagner & Zimmermann, 2017b*). The metrics proposed by them can be used or adapted to determine the adequate granularity of the microservices that are going to be part of an application.

Considering *Bogner, Wagner & Zimmermann (2017b)*, *Candela et al. (2016)* and related papers, we detail the following metrics, which can be used or adapted to define the right granularity of the microservices and to evaluate decompositions.

### Coupling metrics

The coupling measures the degree of dependence of one software component in relation to another. If there is a high degree of coupling, the software component cannot function properly without the other component; furthermore, when we change a software component, we must obligatorily change the other component. For these reasons when designing microservice-based applications, we should look for a low degree of coupling between each microservice.

*Mazlami, Cito & Leitner (2017)* represented the information in the monolith and create an undirected, edge-weighted graph G. Each graph edge has a weight defined by the weight function; this weight function determines how strong the coupling between the neighboring edges is, according to the coupling strategy in use (*Mazlami, Cito & Leitner, 2017*). These coupling strategies can be used as metrics to define the granularity. These metrics are defined as follows:

**Dependency weight.** *Ahmadvand & Ibrahim (2016)* said "dependency weight indicates the frequency of using the dependency. For example, the dependency weight between a billing and shopping cart is high, because with each call to the former a call is required to

**Table 5 Contributions in the definition of the granularity of the microservices by technique, contribution type and automatization level.**

| Paper | Technique | Migration, From Scratch, or None | Input Data | Contribution type | | | | | Automatization level | | |
|---|---|---|---|---|---|---|---|---|---|---|---|
| | | | | Method | Methodology | Ref. Architec. | Problem. | Evaluat. | Manual | Automatic | Semi Auto. |
| P1 | Self-adaptative solution. | None | | | | | X | | X | | |
| P2 | Vertical decomposition in self-contained systems. | Migration | | | | X | | | X | | |
| P3 | Balance cost quality assurance vs deployment. | Migration | | X | | | | | X | | |
| P4 | Comparing the same microservices in a single container and in two containers. | Migration | | | | | | X | X | | |
| P5 | Architecture definition language (ADL). | From Scratch | | | | X | | | X | | |
| P6 | Semantic similarity, clustering k-means, DISCO. | From Scratch | OpenApi specification. | X | | | | | | X | |
| P7 | Domain engineering, domain-driven design. | From Scratch | | X | | | | | X | | |
| P8 | Domain-driven design COSMIC function points. | From Scratch | | X | | | | | X | | |
| P9 | Functional decomposition. | From Scratch | Use cases | X | | | | | X | | |
| P10 | Machine learning method, scale weighted k-means. | Migration | Access logs | X | | | | | | X | |
| P11 | Decomposition form system requirements—security vs scalability | From Scratch | | | X | | | | X | | |
| P12 | Heuristics used for functional splitting, microservice discovery algorithms. | Migration | Source code, database, execution call graphs. | X | | | | | | X | |
| P13 | Graph-based clustering algorithm. | Migration | Source code. | X | | | | | | X | |

 

| Paper | Technique | Migration, From Scratch, or None | Input Data | Contribution type | | | | | Automatization level | | |
|---|---|---|---|---|---|---|---|---|---|---|---|
| | | | | Method | Methodology | Ref. Architec. | Problem. | Evaluat. | Manual | Automatic | Semi Auto. |
| P14 | Dataflow-driven decomposition algorithm. | Migration | Dataflow diagram, uses cases. | | X | | | | | | X |
| P15 | Process-mining approach, DISCO used to identify the business processes. | Migration | Execution logs. | X | | | | | | | X |
| P16 | Search-based functional atom grouping algorithm. Non-dominated sorting genetic algorithm-II (NSGA II). | Migration | Execution traces from logs. | X | | | | | | X | |
| P17 | Virtual machine image synthesis and analysis. | Migration | | | X | | | | X | | |
| P18 | Scheme to refactor SOA services into microservices using decomposition pattern. | From Scratch | Scenario statements, workflow, BPEL description. | | X | | | | X | | |
| P19 | Set of rule-based decisions, adaptation of the four-step rule set (4SRS) method. | From Scratch | UML use cases model. | X | | | | | X | | |
| P20 | Word embedding and hierarchical clustering of semantic similarity. | Migration | OpenApi specification | X | | | | | | X | |
| P21 | Microservice discovery algorithms. Genetic algorithm, DISCO, NSGAII. | Migration | Source code, database, execution call graphs | X | | | | | | | X |
| P22 | A clustering algorithm applied to aggregate domain entities. | Migration | Source code, MVC architectural style. Call graph. | | X | | | | | | X |
| P23 | Service granularity cost analysis-based method, cost analysis function. | From Scratch | | | | | X | | X | | |

| Paper | Technique | Migration, From Scratch, or None | Input Data | Contribution type | | | | | Automatization level | | |
|---|---|---|---|---|---|---|---|---|---|---|---|
| | | | | Method | Methodology | Ref. Architec. | Problem. | Evaluat. | Manual | Automatic | Semi Auto. |
| P24 | Validation framework for microservice decompositions. | Migration | | X | | | | | X | | |
| P25 | Ontology scheme search-based techniques, multi-objective genetic algorithm. | Migration | Component and microservices properties. | | X | | | | | | X |
| P26 | Domain-driven design, architectural design *via* dynamic software visualization. | Migration | None | | X | | | | X | | |
| P27 | Non-dominated sorting genetic algorithm-II (NSGA II). | Migration | Source code | X | | | | | | X | |
| P28 | Fast test model, clustering using affinity propagation algorithm, and clustering of semantically similar API operations. | Migration | OpenAPI specification | X | | | | | | X | |
| P29 | Function call graph, Markov chain model to represent migration characteristics, k-means hierarchical clustering. | Migration | Source code, runtime logs | X | | | | | | | X |

**Note:**
Migration, The proposal is used in migration from monolith to microservices; From Scratch, The proposal can be used for development from scratch; Ref. Architec., reference architecture; Evaluat, evaluation or comparison; Semi Auto, Semi-automatic; DISCO, Distributionally related words using co-occurrences; NSGAII, Non-dominated sorting genetic algorithm-II; SOA, Service-oriented architecture.

the latter. On the other hand, the dependency weight between the billing service and a service managing the meta-data of payment methods is low, because the former calls the latter only once a day" (*Ahmadvand & Ibrahim, 2016*).

**Logical coupling.** *Gall, Jazayeri & Krajewski (2003)* coined the term logical coupling as a retrospective measure of implicit coupling based on the revision history of an application source code (*Gall, Jazayeri & Krajewski, 2003*). *Mazlami, Cito & Leitner (2017)* define the value of the logical coupling is one if classes $(C_1, C_2)$ have changed together in a certain commit. They use the logical coupling aggregate which is the sum of the logical coupling for each pair of classes (*Mazlami, Cito & Leitner, 2017*).

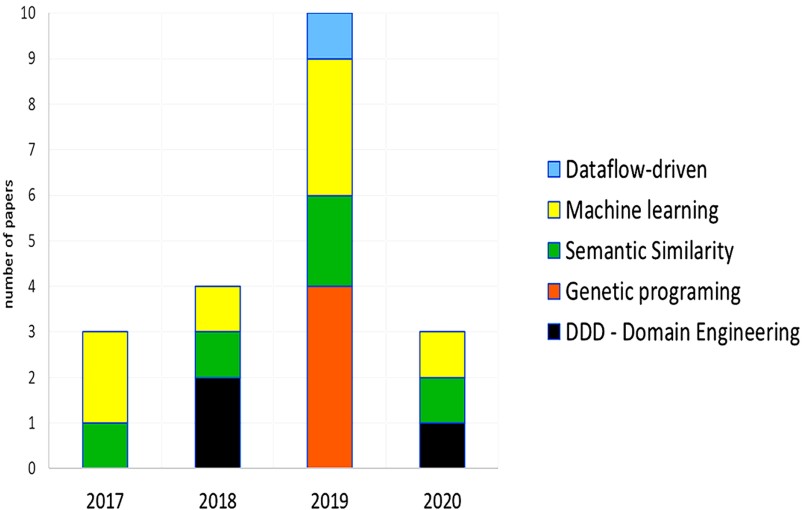

**Figure 7 Type of techniques used in the selected papers.** Multiple techniques may be used by one paper. Some papers used specific techniques, these techniques were not included.

**Semantic coupling.** Basically, semantic coupling couples together classes that contain code about the same things, *i.e.*, domain model entities. The semantic coupling strategy can compute a score that indicates how related two files are, in terms of domain concepts or "things" expressed in code and identifiers (*Mazlami, Cito & Leitner, 2017*).

**Contributor count and contributor coupling.** The contributor coupling strategy aims to incorporate the team-based factors into a formal procedure that can be used to cluster class files according to the team-based factors (reduce communication overhead to external teams and maximize internal communication and cohesion inside developer teams). It does so by analyzing the authors of changes on the class files in the version control history of the monolith. The procedure to compute the contributor coupling is applied to all class files. In the graph $G$ representing the original monolith M, the weight on any edge is equal to the contributor coupling between two classes $C_i$ and $C_j$ that are connected in the graph. The weight is defined as the cardinality of the intersection of the sets of developers that contributed to class $C_i$ and $C_j$ (*Mazlami, Cito & Leitner, 2017*).

**Structural coupling.** Structural coupling consists of the number of classes outside package $P_j$ referenced by classes in the package $P_j$ divided by the number of packages (*Candela et al., 2016*).

**Afferent coupling (Ca).** The number of classes in other packages (services) that depend upon classes within the package (service) itself, as such it indicates the package's (service's) responsibility (*Martin, 2002*) cited by (*Li et al., 2019*).

**Efferent coupling (Ce).** The number of classes in other packages (services), that the classes in a package (service) depend upon, thus indicates its dependence on others (*Martin, 2002*) cited by (*Li et al., 2019*).

**Instability (I).** Measures a package's (service's) resilience to change through calculating the ratio of $Ce$ and $Ce + Ca$. I = 0 indicates a completely stable package (service), whereas I = 1 a completely unstable package (service) (*Martin, 2002*) cited by (*Li et al., 2019*).

**Table 6 Papers and used metrics.**

| Paper | Use of metrics relative to | | | | Metric type | | | | | | | | |
| | Dev. Team | Dev. Process | App. System. | MS. | Coupling | Cohesion | Granularity | Complexity | Team | Comput. Resource. | Source Code | Performance | Other Quality Attributes |
| --- | --- | --- | --- | --- | --- | --- | --- | --- | --- | --- | --- | --- | --- |
| P3 | | X | | | | | | | | | | | Cost of quality assurance. Cost of deployment. |
| P4 | | | | X | | | | | | | | Response time. Number of calls. | |
| P8 | | | X | X | | | | COSMIC function points. | | | | | |
| P10 | | | | X | | | | | | Avg. CPU. Used virtual machines. Allocated virtual machines. | | Response time SLO violations. Number of calls. Number of rejected request. Throughput. | |
| P11 | | | | X | Dependency weight. | | | | | | | | Security impact. Scalability impact. |
| P12 | | | X | X | Structural coupling. | Lack of cohesion. | | | | Avg. Memory. Avg. Disk. | | Number of requests. Execution time. | |
| P13 | X | | X | X | Logical coupling. Average domain redundance. Contributor coupling. Semantic coupling. | | | | Commit count. Contributor count. | | Lines of code | | |
| P14 | | | X | X | Afferent coupling. Efferent coupling. Instability. | Relational cohesion. | | | | | | | |

| Paper | Use of metrics relative to | | | | Metric type | | | | | | | | |
| | Dev. Team | Dev. Process | App. System | MS. | Coupling | Cohesion | Granularity | Complexity | Team | Comput. Resource. | Source Code | Performance | Other Quality Attributes |
|---|---|---|---|---|---|---|---|---|---|---|---|---|---|
| P15 | | | | X | Coupling between microservice. | | | | | | Number of classes. Number of duplicated classes. | | |
| P16 | | | | X | Integrating interface number. | Cohesion at message level. Cohesion at domain level. | | | | | Structural modularity Quality. Conceptual modularity Quality. Internal and external Co-change Frequency (ICF, ECF). Ration of ECF to ICF. | | |
| P21 | | | X | X | Structural coupling. | Lack of cohesion. | | | | Avg. CPU Avg. Network. | | Number of executions. Number of packets send. | |
| P22 | | | X | | Silhouette Score. | | | Number of singleton clusters. Maximum cluster size. | | | | | |
| P23 | | | | | | | | | | | Service composition cost. Service decomposition cost. | Request time | |
| P24 | | | X | X | Dependencies composition. Strongly connected components. | Nano-entities composition. Relation composition. Responsibilities composition. Semantic similarity. | Number of nano-entities. | | | | | | |

(Continued)

| Paper | Use of metrics relative to | | | Metric type | | | | | | | | |
| | Dev. Team | Dev. Process | App. System. | MS. | Coupling | Cohesion | Granularity | Complexity | Team | Comput. Resource. | Source Code | Performance | Other Quality Attributes |
|---|---|---|---|---|---|---|---|---|---|---|---|---|---|
| P27 | | | | X | | Cohesion at Message level. Cohesion at Domain level. | Operation number. | | | | | Interaction number. | |
| P28 | | | X | X | | Lack of cohesion. | Operation number. | | | | | | |
| P29 | | | X | | | | Number of microservices. Number of interfaces. | | | | Lines of code. Package analysis. Class hierarchy analysis. | Static structure analysis. Static call graph analysis. Combined static and dynamic analysis. | |

**Note:**

Paper, paper number; Dev. Team, Development team; Dev. Process, Development process; App. System, Application or system; MS, Microservice; Comput. Resource, Computational resource; Avg, Average; COSMIC, Common Software Measurement International Consortium; SLO, Service-level objective.

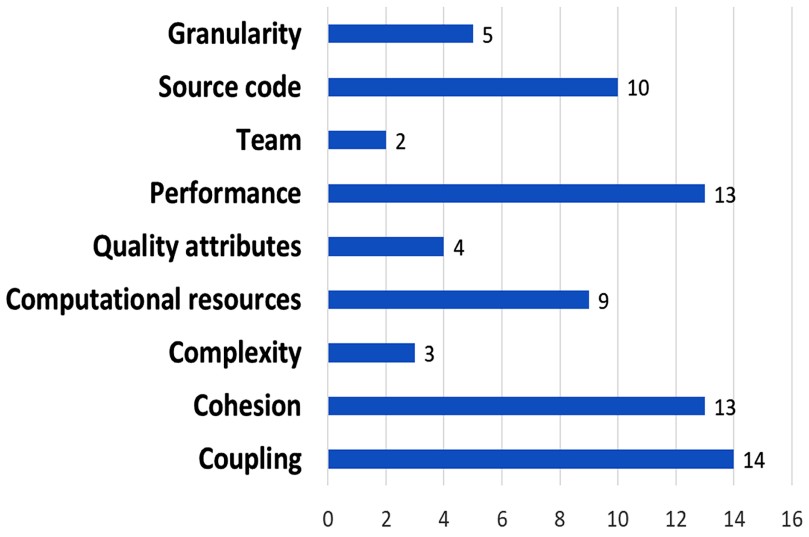

**Figure 8** Amount of metrics per metric type.   

**Coupling between microservice (CBM).** *Taibi & Syst (2019)* defined CBM extending the well-known coupling between objects; this coupling can occur through method calls, field accesses, inheritance, arguments, return types, and exceptions. They calculated the relative CBM for each microservice by dividing the number of external links between the number of classes in the microservices (*Taibi & Syst, 2019*).

**Integrating interface number (IFN).** The number of published interfaces of a service. The smaller the IFN, the more likely the service is to assume a single responsibility. The IFN for the system is the average of all IFN (*Jin et al., 2019*).

**Average domain redundancy (ADR).** A favorable microservice design avoids duplication of responsibilities across services. *Mazlami, Cito & Leitner (2017)* computed the average domain redundancy metric as a proxy to indicate the amount of domain-specific duplication or redundancy between the services on a normalized scale between 0 and 1, where we favor service recommendations with lower ADR values (*Mazlami, Cito & Leitner, 2017*).

**Absolute importance of the service (AIS).** The number of clients that invoke at least one operation of a service's interface (*Rud, Schmietendorf & Dumke, 2006*).

**Absolute Dependence of the Service (ADS).** The number of other services that service S depends on, *i.e.*, the number of services from which S invokes at least one operation (*Rud, Schmietendorf & Dumke, 2006*).

**Services interdependence in the system (SIY).** The number of service pairs that are bi-directionally dependent on each other (*Rud, Schmietendorf & Dumke, 2006*).

**Dependencies composition.** *Cojocaru, Uta & Oprescu (2019)* defined dependencies composition as "The test assesses how balanced outward dependencies are across the microservices, by counting the outward dependencies each microservice has toward its peers. The algorithm constructs a dependency graph of the system where each dependency represents a communication path utilized for exchanging data between two components of the system" (*Cojocaru, Uta & Oprescu, 2019*).

**Strongly connected components (SCC).** SCC identifies strongly connected components on the previously mentioned dependency graph using Tarjan's algorithm. If cycles are detected in the communication paths, then the respective services should be aggregated into one microservice. The score is divided by the total number of microservices within the system (*Cojocaru, Uta & Oprescu, 2019*).

**Silhouette score.** *Nunes, Santos & Rito Silva (2019)* define the silhouette score as "the difference between the mean nearest-cluster distance $a$ and the mean intra-cluster distance $b$ divided by the greatest of $a$ and $b$. This score ranges its values from −1 to 1, representing incorrect clustering (samples on wrong clusters) and highly dense clustering, respectively. This metric creates a parallelism with the overall coupling of the clusters of the system, as our objective was to obtain a high intra-cluster similarity and a low inter-cluster similarity, so the partition between clusters is well defined" (*Nunes, Santos & Rito Silva, 2019*).

*Perepletchikov et al. (2007)* proposed a set of design-level metrics to measure the structural attribute of coupling in service-oriented systems, which can be adapted to microservices (*Perepletchikov et al., 2007*).

### Cohesion metrics

Cohesion and coupling are two contrasting goals. A solution balancing high cohesion and low coupling is the goal for developers. *Candela et al. (2016)* employed a two-objective approach aimed at maximizing package cohesion and minimizing package coupling. They used class dependencies and structural information to measure the structural cohesion, which can be adapted to microservices. The metrics used in the related work are listed below:

**Lack of cohesion.** Lack of cohesion of classes for the $J_{th}$ package $(P_j)$ measured as the number of pairs of classes in $P_j$ with no dependency between them (*Candela et al., 2016*). *Al-Debagy & Martinek (2020)* used the metric in a different way as "It is based on Henderson-Sellers's lack of cohesion metrics (*Al-Debagy & Martinek, 2020*). But the proposed version is modified to be applicable for microservices' APIs. It works by finding how many times a microservice has used a specific operation's parameter, divided by the product of the number of operations multiplied by the number of unique parameters". *Candela et al. (2016)* used lack of structural cohesion to measure the lack of cohesion between classes divided by the number of packages, and they also defined a lack of conceptual cohesion as a metric (*Candela et al., 2016*).

**Relational cohesion.** Relational cohesion was defined as "the ratio between the number of internal relations and the number of types in a package (service). Internal relations include inheritance between classes, invocation of methods, access to class attributes, and explicit references like creating a class instance. Higher numbers of RC indicate higher cohesion of a package (service)" (*Larman, 2004*) cite by (*Li et al., 2019*).

**Cohesion at the domain level (CHD).** The cohesiveness of interfaces provided by a service at the domain level. The higher the CHD, the more functionally cohesive that service is (*Jin et al., 2019*).

**Average cohesion at the domain level (Avg. CHD).** The average of all CHD values within the system (*Jin et al., 2019*).

**Cohesion at the message level (CHM).** The cohesiveness of the interfaces published by a service at the message level. The higher a service's CHM, the more cohesive the service is, from an external perspective. CHM is the average functional cohesiveness (*Jin et al., 2019*).

**Service interface data cohesion (SIDC).** The cohesion of a given service *S* with respect to the similarity of parameter data types of its interface's operations (*Perepletchikov, Ryan & Frampton, 2007*).

**Service Interface Usage Cohesion (SIUC).** The cohesion of a given service *S* based on the invocation behavior of clients using operations from its interface (*Perepletchikov, Ryan & Frampton, 2007*).

**Entities composition.** According to *Cojocaru, Uta & Oprescu (2019)*, "entities composition assesses whether the entities are equally distributed among the proposed microservices and no duplicates, which might break the cohesion, exist. They define an entity as the class, or action of the service" (*Cojocaru, Uta & Oprescu, 2019*).

**Relation composition.** According to *Cojocaru, Uta & Oprescu (2019)* "relation composition assesses the quantitative variation in published language per relation. It applies the concept of relative assessment to entities shared between the services *via* their communication paths. The test identifies services communicating much more data than their peers, and thus potential communication bottlenecks" (*Cojocaru, Uta & Oprescu, 2019*).

**Responsibilities composition.** *Cojocaru, Uta & Oprescu (2019)* stated that the responsibilities composition "assesses to what extent the use case responsibilities are equally distributed among the proposed microservices. It uses the coefficient of variation between the number of use case responsibilities of each microservice. Services having relatively more responsibility may imply low cohesion: a service providing multiple actions violates the single responsibility principle" (*Cojocaru, Uta & Oprescu, 2019*).

**Semantic similarity:** According to *Cojocaru, Uta & Oprescu (2019)* "semantic similarity uses lexical distance assessment algorithms to flag the services that contain unrelated components or unrelated actions hindering cohesion" (*Cojocaru, Uta & Oprescu, 2019*).

*Perepletchikov, Ryan & Frampton (2007)* "reviewed categories of cohesion initially proposed for object-oriented software in order to determine their conceptual relevance to service-oriented designs"; and proposed a set of metrics for cohesion that can be adapted for microservices (*Perepletchikov, Ryan & Frampton, 2007*).

### Complexity metrics

The complexity of microservices should be low, so that they can be changed within several weeks, rewritten, and improved quickly. If the complexity is high, then the cost of change is higher. Measuring complexity is fundamental for developing microservice-based applications. The metrics used by the authors of the papers are listed below.

**Function points.** A method for measuring the size of a software. A function point count is a measurement of the amount of functionality that software will provide (*Totalmetrics. com, 2020*).

**COSMIC function points.** Focuses on data movements between different layers. One of the benefits of the COSMIC method is that it can estimate the size in the planning phase, based on the user's functional requirements. The four main data group types are: entry, exit, read and write. The COSMIC function point calculation is aimed at measuring the system at the time of planning. This size calculation can be used for estimating efforts (*Vural, Koyuncu & Misra, 2018*).

**Total response for service (TRS).** The sum of all responses for operation (RFO) values for all operations of the interface of service S (*Perepletchikov et al., 2007*).

**Number of singleton clusters (NSC).** *Nunes, Santos & Rito Silva (2019)* said "that having more than two singleton clusters is considered negative. Considering a final microservice architecture with clear functional boundaries established, it is likely that there are not two services in which their content is a single domain entity" (*Nunes, Santos & Rito Silva, 2019*).

**Maximum cluster size (MCS).** *Nunes, Santos & Rito Silva (2019)* "MCS should not be bigger than half of the size of the system. Even with a cluster size inside this range, there is also a dependency regarding the number of entity instances that are part of the aggregate" (*Nunes, Santos & Rito Silva, 2019*).

### Performance metrics

Performance is a critical point of microservice-based applications. The selected papers used eight performance metrics:

**The number of calls or requests.** The number of times that a microservice is called.

**The number of rejected requests.** The number of times that a microservice does not respond or exceeds the time limit.

**Response or execution time.** The execution time of the invoked service.

**Interaction number (IRN).** The number of calls for methods among all pairs of extracted microservices. The smaller is the IRN, the better the quality of candidate microservices as a low IRN reflects loose coupling (*Saidani et al., 2019*).

**Number of executions.** The number of test requests sent to the system or microservices.

**Maximum request time.** The maximum time for a request (output) made from one microservice to another.

**Maximum response time:** The maximum response time is that of a call (input) or request to the system or microservice. It is the time to process a response to another microservice.

**Number of packets sent:** The packets sent to the system or microservice.

*Ren et al. (2018)* (P29) they used package analysis (PA), static structure analysis (SSA), class hierarchy analysis (CHA), static call graph analysis (SCGA), and combined static and dynamic analysis (CSDA) to evaluate migration performance. However, they did not explain the details of the performance analysis test or the metrics they used (*Ren et al., 2018*).

### Other quality attributes metrics

Few metrics were directly related to quality attributes. The metrics proposed in the revised works are defined as follows:

**Cost of quality assurance.** It can be calculated by adding up the time spent by testers validating not only the new features but also the non-regression on existing ones, along with the time spent on release management (*Gouigoux & Tamzalit, 2017*).

**Cost of deployment.** The time spent by operational teams to deploy a new release, in man-days; it decreases greatly as teams automate deployment (*Gouigoux & Tamzalit, 2017*).

**Security impact.** The security policy applied to requirements or services. Assets and threats identified lead to deployed security mechanisms, which form security policies. *Ahmadvand & Ibrahim (2016)* mapped the identified policies to the corresponding functional requirements, mainly based on their access to the system assets. Security impact is a qualitative value (low, medium, high) (*Ahmadvand & Ibrahim, 2016*).

**Scalability impact.** *Ahmadvand & Ibrahim (2016)* define the Scalability impact as the required level of scalability (high, medium, low) to implement a a functional requirement or service. Defining the requirements at design time for a software system to be scalable is a challenging task. *Ahmadvand & Ibrahim (2016)* think that a requirements engineer should answer a question such as "What is the anticipated number of simultaneous users for this functionality?" (*Ahmadvand & Ibrahim, 2016*).

### Computational resources metrics

The computational resources are all the software and hardware elements necessary for the operation of the microservice-based applications. The proposed metrics are listed.

**Average of memory.** The average memory consumption for each microservice or application.

**Average of disk.** The average of disk consumption for each microservice or application.

**Average of network.** The average network bandwidth consumption for the entire system; Kb/s used by system or microservice (*De Alwis et al., 2019*).

**Average of CPU.** The average of the CPU consumption by the system or microservice.

**Service composition cost (SCC).** *Homay et al. (2019)* stated that "identifying which existing functionalities in the service are consuming more resources is not an easy task. Therefore, we suggest relying on each request that a service provider receives from a service consumer. Because each request is a chain of stats or activities that needs to be satisfied inside of the service provider to generate a related response. The cost-of-service composition for the service s will be equal to the maximum cost of requests (routes)" (*Homay et al., 2019*).

**Service decomposition cost (SDC).** According to *Homay et al. (2019)*, "By refining a service into smaller services, we will make some drawbacks. The SDC is a function that calculates the overhead of refining the service s into smaller pieces" (*Homay et al., 2019*).

### Team metrics

Each microservice can be developed by a different team, and with different programming languages and database engines. It is important to consider metrics that allow analysis of microservices' granularity and its impacts on the development team. The proposed metrics in the analyzed are as follows:

**Team size reduction (TSR).** Reduced team size translates to reduced communication overhead and thus more productivity and the team's focus can be directed toward the actual domain problem and service for which it is responsible. TSR is a proxy for this team-oriented quality aspect. Let *RM* be a microservice recommendation for a monolith *M*. *TSR* is computed as the average team size across all microservice candidates in the *RM* divided by the team size of the original monolith *M* (*Mazlami, Cito & Leitner, 2017*).

**Commit count.** The number of commits in the code repository made by the developers.

We found very few metrics related to the development team. This can be an interesting topic for future research.

### Source code metrics

The source code is one of the most important sources for analyzing certain characteristics of an application. Some authors have used it to identify microservices and define their granularity. The proposed metrics are described below:

**Code size in lines of code.** The total size of the code in the repository, in terms of in lines of code, microservices' lines of code, or application's lines of code.

**The number of classes per microservice.** Helps to understand how large the identified microservice is and to identify if any microservice is too big compared to others. The number of classes should be minimized because a smaller number of classes implies more independent development of the microservice (*Taibi & Syst, 2019*).

**The number of duplicated classes.** In some cases, two execution traces will have several classes in common. The number of duplicated classes helps one to reason about the different slicing options, considering not only the size of the microservices but also the number of duplications, that will be then reflected in the microservices' development. Duplicated classes should be avoided since duplication adds to the system's size and maintenance (*Taibi & Syst, 2019*).

**Internal co-change frequency (ICF).** How often entities within a service change together as recorded in the revision history. A higher ICF means that the entities within this service will be more likely to evolve together. The ICF is the average of all ICFs within the system (*Jin et al., 2019*).

**External co-change frequency (ECF).** How often entities assigned to different services change together, according to the revision history. A lower ECF score means that entity pairs located in different services are expected to evolve more independently. Similarly, ECF is the average ECF value of all services within the system (*Jin et al., 2019*).

**The ratio of ECF to ICF (REI).** The ratio of co-change frequency across services *vs.* the co-change frequency within services. The ratio is expected to be less than 1.0 if co-changes happen more often inside a service than across different services. The smaller the ratio is, the less likely co-changes are across services, and the extracted services tend to

evolve more independently. Ideally, all co-changes should happen inside the services. REI is calculated as ECF divided by ICF (*Jin et al., 2019*).

**Modularity quality measure.** The modularity of a component or service can be measured from multiple perspectives, such as structural, conceptual, historical, and dynamic dimensions(*Candela et al., 2016*). They extend the modularity quality (MQ), as defined by *Mancoridis et al. (1998)* as structural and conceptual dependencies, using structural modularity quality and conceptual modularity quality to assess the modularity of service candidates. Structural modularity quality (SMQ) measures the quality of modularity from a structural perspective. The higher the SMQ, the better modularized the service is. On the other hand, conceptual modularity quality (CMQ), similarly measures modularity quality from a conceptual perspective. The higher the CMQ, the better (*Jin et al., 2019*).

### Granularity metrics

Measuring granularity is complex. Granularity is related to size, including the number of functionalities or services that the application or microservice will have. It is also related to coupling and cohesion. Being more granular implies that microservice has no dependencies and can function independently, as an independent and encapsulated piece. Six granularity metrics were identified:

**Weighted service interface count (WSIC):** WSIC(S) is the number of exposed interface operations of service *S*. The default weight is set to one. Alternate weighting methods, which need to be validated empirically, can take into consideration the number and the complexity of data types of parameters in each interface (*Hirzalla, Cleland-Huang & Arsanjani, 2009*). WSIC(S) is the number of exposed interface operations of service S. Operations can be weighted based on the number of parameters or their granularity (*e.g.*, a complex nested object) with the default weight being set to one (*Bogner, Wagner & Zimmermann, 2017a*).

**Component Balance (CB):** The CB is a system-level metric to evaluate the appropriateness of granularity, *i.e.*, if the number and size uniformity of the components (in this case, services) are in a favorable range for maintainability (*Bouwers et al., 2011*).

**Operation number (OPN):** The OPN is used to compute the average number of public operations exposed by an extracted microservice to other candidate microservices. The smaller the OPN is the better (*Saidani et al., 2019*).

**Number of microservices:** The number of microservices that are part of the system or application.

**Lines of code:** The lines of code measure the number of lines of code in the microservice. Additionally, it may consider the total size of the code in the repository.

**Number of nanoentities:** *Cojocaru, Uta & Oprescu (2019)* stated that "the number of nanoentities (attributes or fields of a class) computes the number of nanoentities assigned to each proposed service, storing the result as a floating-point parameterized list. The list's length is equal to the number of services found in the system model specification file" (*Cojocaru, Uta & Oprescu, 2019*).

The fundamentals of microservices suggest that they must have low coupling, high cohesion, and low complexity. Based on the described metrics, a model or method could be defined that uses artificial intelligence to determine the most appropriate dimensioning and size for microservices. Some have already been defined, mainly for migrations from monoliths to microservices. The number of microservices, their size, and their computational complexity directly affect the use of computational resources and therefore their cost of deployment. This is an interesting topic for future research.

In conclusion, some papers used metrics to evaluate the granularity of the microservices, including coupling, cohesion, number of calls, number of requests, and response time, although few methods or techniques use complexity as a metric, even though it seems fundamental for microservices. More research that considers design-level metrics is needed to define the granularity of the microservices that are part of an application, as well as research proposing models, methods, or techniques to determine the most appropriate granularity.

## RQ3: Quality attributes to define the microservice granularity

Quality attributes are essential for today's applications. Availability, performance, automatic scaling, maintainability, security, and fault tolerance are essential features that every application must handle. An architecture based on microservices allows independent management of quality attributes, according to the specific need of each microservice. This is one of the main advantages compared to monolithic architectures.

The size and number of microservices that compose an application directly affect its quality attributes. Creating more microservices may affect maintainability because testing costs will increase, even more so if automated testing is not available. Moreover, performance may also be affected by having to integrate and process data from several distributed applications. Clearly, quality attributes are impacted by microservices granularity and should be considered when defining a model, method, or technique to determine granularity (see Fig. 9).

Surprisingly, 62% of the identified proposals did not consider or report any quality attributes at all. Of those that did, scalability and performance were the most considered (seven papers, 24%), followed by maintainability and availability (two papers, 7%); and lastly, reliability (fault tolerance), security, functionality, and modularity with only one paper each. More research is needed that considers quality attributes to define the granularity of the microservices comprising an application. Security and fault tolerance are key attributes that microservice-based applications must handle, few works addressed these features (see Table 7).

We grouped the software quality attributes into the following two categories: firstly, according to runtime characteristics, (scalability, performance, reliability, availability, and functionality), which are observable during execution; and secondly according to software as an artifact characteristic (maintainability, modularity, reusability), which are not observable during execution (*Bass, Clemens & Katzman, 1998*; *Astudillo, 2005*); run time characteristics were the most used ones, having been addressed by eight papers (P2, P4, P10, P11, P12, P13, P21 y P29); only two papers addressed software artifact characteristics

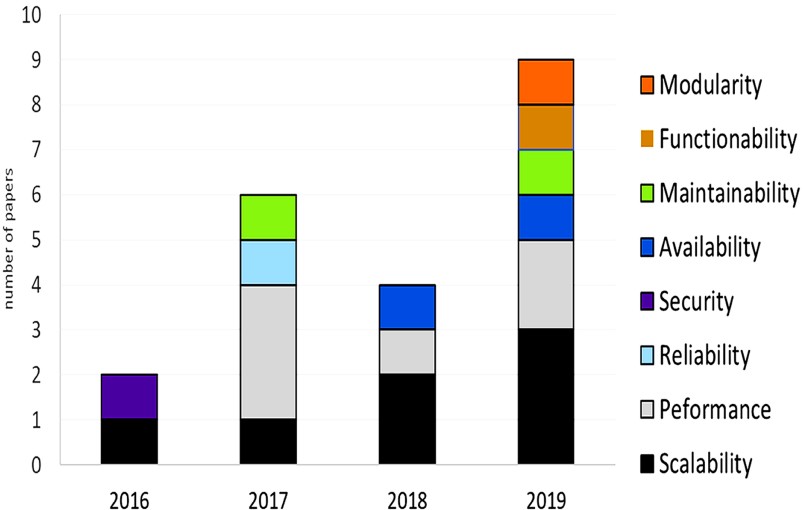

**Figure 9 Amount of papers per addressed quality attributes and year.**

**Table 7 Papers and quality attributes grouped by runtime and artifact software characteristics.**

| Paper | Runtime characteristics | | | | | | Software artifact characteristics | |
|---|---|---|---|---|---|---|---|---|
| | Scalab. | Perfor. | Reli. | Avail. | Security | Funct. | Maint. | Modul. |
| P2 | X | X | X | | | | | |
| P3 | | | | | | | X | |
| P4 | | X | | | | | | |
| P10 | X | X | | | | | | |
| P11 | X | | | | X | | | |
| P12 | X | X | | X | | | | |
| P13 | | X | | | | | | |
| P16 | X | | | | | X | X | X |
| P21 | X | X | | X | | | | |
| P29 | X | X | | | | | | |

**Note:**
Scalab, Scalability; Perfor, Performance; Reli, Reliability; Avail, Availability; Funct, functionality; Main, Maintainability; Evol, Evolvability; Modul, Modularity.

(P3 y P16); only one paper used both artifact and runtime characteristics (P16). Therefore, more proposals are required to define microservices granularity considering both runtime and software artifact characteristics.

### Runtime characteristics

In this section, we detail the runtime quality attributes and the way they were addressed by the papers, whether they used metrics to evaluate their proposals.

- Scalability, performance, and reliability (fault tolerance) were used by only one paper (P2). P2 proposed a re-implementation of otoo.de (a real-life case study). They defined the granularity through vertical decomposition, they used DevOps including continuous deployment, to deliver features quickly to customers. Team organization is

crucial for success, this organization was based on Conway's Law. Full automation of quality assurance and software deployment allows for early fault and error detection, thus reducing repair times both during development and during operations (*Hasselbring & Steinacker, 2017*). This paper did not propose metrics for evaluation.

- Scalability and performance were used by two papers (P10, P29); P10 proposed an automatic decomposition method, which was based on a black-box approach that mines the application access logs using a clustering method to discover URL partitions having similar performance and resource requirements. Such partitions were mapped to microservices (*Ahmadvand & Ibrahim, 2016*). The metrics used in this paper were performance metrics (response time SLO violations, number of calls, number of rejected requests, and throughput) and computational resource metrics (Avg. CPU, number of virtual machines used, and allocated virtual machines). P29 used the source code and the runtime logs in a semi-automatic method, it used granularity, performance, and source code metrics to evaluate the decompositions. They presented a program analysis-based method to migrate monolith legacy applications to microservices architecture; this method used a function call graph, a Markov chain model to represent migration characteristics, and a k-means hierarchical clustering algorithm (*Ren et al., 2018*).

- Only performance was used by two papers (P4, P13). P4 examined the granularity problem of the microservice and explored its effect on the latency of the application. Two approaches for the deployment of microservices were simulated; the first one with microservices in a single container, and the second one with microservices divided into separate containers. They discussed the findings in the context of the Internet of Things (IoT) application architectures (*Shadija, Rezai & Hill, 2017*); that paper corresponds to an evaluation or comparison, it is not a method to define the microservice granularity; it used performance metrics (response time and the number of calls). P13 presented three formal coupling strategies and embedded those in a graph-based clustering algorithm: (1) logical coupling, (2) semantic coupling, and (3) contributor coupling. The coupling strategies rely on meta-information from monolithic code bases to construct a graph representation of the monoliths that are in turn processed by the clustering algorithm to generate recommendations for potential microservice candidates in a refactoring scenario; P13 was the only one that proposed development team based metrics; logical coupling, average domain redundancy, contributor coupling, semantic coupling, commit count, contributor count, and lines of code were the metrics used by this paper.

- Scalability and security were used by one paper (P11). P11 proposed a methodology, consisting of a series of steps and activities that must be carried out to identify the microservices that will be part of the system. It is based on the use cases and the analysis made by the architect in terms of the scalability and security of each use case, as well as the dependencies with the other use cases (*Ahmadvand & Ibrahim, 2016*); this paper used the following metrics: dependency weight, security impact, and scalability impact (qualitative metrics).

- Scalability, performance, and availability were addressed by two papers (P12, P21). P12 presented discovery techniques that help identifying the appropriate parts of consumer-oriented business systems that could be redesigned as microservices with desired characteristics such as high cohesion, low coupling, high scalability, high availability, and high processing efficiency (*De Alwis et al., 2018*). They proposed microservice discovery algorithms and heuristics. It was an automatic method that used coupling (structural coupling), cohesion (lack of cohesion), computational resources (avg. memory, avg. disk), and performance metrics (number of requests, execution time). P21 was a semi-automatic method, a genetic algorithm with semantic similarity based on DISCO and non-dominated sorting genetic algorithm-II (NSGAII). That paper presented four microservice patterns, namely object association, exclusive containment, inclusive containment, and subtyping for 'greenfield' (new) development of software while demonstrating the value of the patterns for 'brownfield' (evolving) developments by identifying prospective microservices (*De Alwis et al., 2019*).
  The metrics used by that paper were: structural coupling, lack of cohesion, the average CPU, the average of the network, number of executions, and the number of packets sent.

### Software as an artifact characteristic

Only two software as an artifact characteristic were used, which were maintainability and modularity. Only maintainability was used by one paper (P3); whereas maintainability, and modularity were used by (P16), which proposed the most complete method.

P3 used a balance between the cost of quality assurance and the cost of deployment for defining microservices granularity, it was a manual method. The choice of granularity should be based on the balance between the costs of quality assurance and the cost of deployment (*Gouigoux & Tamzalit, 2017*).

P16 presented a framework that consists of three major steps: (1) extracting representative execution traces, (2) identifying entities using a search-based functional atom grouping algorithm, and (3) identifying interfaces for service candidates (*Jin et al., 2019*). They also presented a comprehensive measurement system to quantitatively evaluate service candidate quality in terms of functionality, modularity, and evolvability. P16 proposed an automatic method, which used a search-based functional atom grouping algorithm and a non-dominated sorting genetic algorithm-II (NSGA II).
The evaluation metrics were coupling (Integrating interface number), cohesion at message level, cohesion at domain level, structural modularity quality, conceptual modularity quality, internal co-change frequency (ICF), external Co-change frequency (ECF), and the ratio of ECF to ICF.

### Quality attributes and artificial intelligence

In some cases, artificial intelligence techniques are being used to improve the quality attributes of microservices. For example:

*Alipour & Liu (2017)* proposed two machine learning algorithms and predicted the resource demand of microservice backend systems, as emulated by a Netflix workload reference application. They proposed a microservice architecture that encapsulates

monitoring functions of metrics and learning of workload patterns. Then, this service architecture is used to predict the future workload for making decisions about resource provisioning (*Alipour & Liu, 2017*).

*Prachitmutita et al. (2018)* proposed a new self-scaling framework based on the predicted workload, with an artificial neural network, a recurrent neural network, and a resource scaling optimization algorithm used to create an automated system to manage the entire application with Infrastructure-as-a-service (IaaS) (*Prachitmutita et al., 2018*).

*Ma et al. (2018)* proposed an approach, called scenario-based microservice retrieval (SMSR), to recommend appropriate microservices for users based on the Behavior-driven Development (BDD) test scenarios written by the user. The proposed service retrieval algorithm is based on word2vec, an automatic learning method widely used in natural language processing (NLP) to perform service filtering and calculate service similarity (*Ma et al., 2018*).

*Abdullah, Iqbal & Erradi (2019)* proposed a complete automated system for breaking down an application into microservices, implementing microservices using appropriate resources, and automatically scaling microservices to maintain the desired response time (*Abdullah, Iqbal & Erradi, 2019*).

Artificial intelligence can help to improve and control different characteristics of microservices, especially those related to improving quality attributes. Some proposals have been made in this regard, but more research is needed.

Finally, we identified the automatic and semi-automatic methods, which used metrics and addressed some quality attribute to define the granularity (see Table 8). Only six papers meet those conditions (P10, P12, P13, P16, P21, and P29), and were the more suitable methods to define the granularity of microservices.

Also, we identified semi-automatic methodologies, which used metrics to define the granularity, only two papers were found (P14 and P22); there were no automatic methodologies, most of them were manual methodologies (P11, P17, P18, and P26) and only one was semi-automatic but it did not use metrics (P25).

P14 was a data flow-driven decomposition algorithm. Their methodology, first, the use case specification and business logics were analyzed based on requirements; second, the detailed dataflow diagrams (DFD) at different levels and the corresponding process-datastore version of DFD (DFD PS) are constructed from business logics based on requirement analysis; third, we designed an algorithm to automatically condense the DFD PS to a decomposable DFD, in which the sentences between processes and data stores are combined; last but not least, microservice candidates were identified and extracted automatically from the decomposable DFD (*Li et al., 2019*). The metrics used by P14 were coupling (afferent coupling, efferent coupling, instability) and cohesion (relational cohesion) metrics. P14 did not address any quality attribute directly.

P22 proposed a clustering algorithm applied to aggregate domain entities. It used coupling (silhouette score) and complexity (number of singleton clusters, maximum cluster size) metrics. The authors proposed an approach to the migration of monolith applications to a microservice architecture that focused on the impact of the decomposition on the monolith business logic (*Nunes, Santos & Rito Silva, 2019*).

**Table 8 Semi-automatic methods, which used metrics and addressed some quality attribute to define the granularity.**

| Paper | Metrics | Quality attributes |
|---|---|---|
| P10 | Response time.<br>Number of calls.<br>Number of rejected request. | Scalability and performance |
| P12 | Structural coupling.<br>Lack of cohesion.<br>Avg. Memory.<br>Avg. Disk.<br>Number of requests.<br>Execution time. | Scalability, performance, and availability |
| P13 | Logical coupling.<br>Average domain redundance.<br>Contributor coupling.<br>Semantic coupling.<br>Commit count.<br>Contributor count.<br>Lines of code | Performance |
| P16 | Integrating interface number.<br>Cohesion at message level.<br>Cohesion at domain level.<br>Structural modularity Quality.<br>Conceptual modularity Quality.<br>Internal and extermal Co-change Frequency (ICF, ECF).<br>Ration of ECF to ICF. | Scalability, functionality, maintainability, evolvability, and modularity |
| P21 | Structural coupling.<br>Lack of cohesion.<br>Avg. CPU.<br>Avg. Network.<br>Number of executions.<br>Number of packets send. | Scalability, performance, and availability |
| P29 | Number of microservices.<br>Number of interfaces.<br>Lines of code.<br>Package analysis.<br>Class hierarchy analysis.<br>Static structure analysis.<br>Static call graph analysis.<br>Combined static and dynamic analysis. | Scalability and performance. |

# DISCUSSION

The use of artificial intelligence techniques to determine the appropriate granularity or to identify the microservices that will be part of an application is a growing trend; this is especially true of machine learning clustering algorithms and genetic algorithms, with an emphasis on semantic similarity to group the microservices that refer to the same entity. Domain engineering and DDD are still among the most used techniques.

Migration of software systems implies many architectural decisions that should be systematically evaluated to assess concrete trade-offs and risks (*Cruz et al., 2019*). In these cases, the beginning is given from a monolithic system that must be decomposed into microservices, and that monolithic system has important data sources that allow for the

identification and evaluation of the candidate microservices. These sources are mainly: the source code, the use cases, the database, the logs, and the execution traces. It should be noted that the development of microservice-based applications is closely related to agile practices and DevOps, and none of the input data that are being considered in the proposed methods correspond to agile artifacts such as user stories, product backlog, iteration planning and others. Therefore, research work is needed at this point.

The migration from monolith to microservices is a topic with much interest and is widely studied. In contrast, the design and development of microservice-based applications from scratch have few related proposals. The proposed methods emphasize artifacts available at run time, development, deployment, or production, which are hardly available when starting a project from scratch at design time.

The development of microservice-based applications from scratch resembles component-based development (*Vera-Rivera & Rojas Morales, 2010*), in which microservices are reusable software components. In *Vera-Rivera (2018)*, we characterized the process of developing applications based on microservices, identifying two fundamental parts, first the development of each microservice and then the development of applications based on those microservices.

The definition of adequate granularity is fundamental to the development of microservices-based applications (*Vera-Rivera, 2018*). The granularity of a monolith is not the most optimal and defining an operation by microservice is also not optimal. Hence, if an application that offers 100 operations should have 100 microservices, it would not be optimal either, due to latency, performance, and management of this large distributed system. The optimal granularity is somewhere in between the monolithic application and the operation by microservice system, this granularity should be defined according to the characteristics of the application, the development team, the non-functional requirements, the available resources, and design, development, and operation trade-offs.

The research gaps focus on proposing techniques or methods that allow for the evaluation of granularity and its impact on tests, considering security controls, fault tolerance mechanisms and DevOps. By managing more microservices or larger microservices, testing can be slower and more tedious. Moreover, the pipelines of continuous integration and deployment would be more complex. Determining the appropriate number of microservices and their impact on continuous deployment is an interesting research topic. Few works address these issues.

In addition, few papers use as input data or analysis units the artifacts used in agile development, such as user stories, product backlog, release planning, Kanban board and its data, to propose agile methods or new practices that allow for determination or evaluation of the microservices that will be part of the application. None of the proposed works focuses on agile software development.

Several interesting works have been proposed, but there are still few specific, actionable proposals; more research is needed to propose design patterns, good practices, more complete models, methods, or tools that can be generalized to define microservices granularity considering metrics, quality attributes and trade-offs.

## Research trends

We detail the research trends according to the analyzed papers, the trends are summarized as follows:

- The most used techniques to define microservice granularity included machine learning clustering, semantic similarity, genetic programming, and domain engineering.
- The most used research strategies were validation research and solution proposals.
- The most used validation method was the case study, although some studies used experimental evaluations. We summarized the case studies found in the reviewed papers because they are valuable resources with which to validate future research and to compare new methods (see Table 4). The most common case studies were Kanban boards, Money transfer, JPetsStore and Cargo Tracking, which are either hypothetical or open-source projects.
- The use of metrics was evidenced to evaluate the granularity of the microservices comprising an application. Performance and coupling were the most used metrics; they help to identify microservices and their granularity more objectively.
- Migrations from monoliths to microservices have been widely studied. Methods and techniques have been proposed to decompose applications into microservices, with the source code, logs, execution traces, and even use cases used as input data. These methods are used mainly during design and development time.
- Scalability and performance were the most addressed quality attributes in the reviewed papers; they are fundamental for microservice-based applications. Finally, the main reason to migrate a monolithic application to microservices is precisely to improve performance and scalability, followed by fault tolerance, maintainability, and modularity.

## Research gaps

Research gaps allow us to propose new research works and future work, we identify the research gaps, which are listed as follow:

- Research works that include techniques or methods to evaluate granularity and its impact on tests, while also considering security controls, fault-tolerance mechanisms, and DevOps.
- Metrics were grouped into four categories: development team, development process, microservice-based application (system), and microservice itself. Few metrics were found for the development team or development process, more research is necessary in these groups.
- Few methods have been proposed to define the most adequate microservices granularity at testing or deployment time.
- More research is required that uses agile development artifacts as inputs, (*i.e.,* user stories, product backlog, release planning, Kanban boards, and their data), to propose

new agile practices to define or assess microservices' granularity. None of the proposal identified in this survey focused on agile software development.

## THREATS TO VALIDITY

### External validity

We express a threat to external validity regarding the search and selection of primary papers, which may not be representative of the state of the art in the definition of the granularity of microservices, to reduce this risk we used a systematic and well defined process, using two search strings so that the papers obtained are representative, to define and select the papers that were included in our review, each of the authors made their selection and tabulation independently, then in common agreement and group discussion were selected by applying the inclusion and exclusion criteria. In addition, the systematic literature review process carried out corresponds to the classic and widely used in other reviews, proposed by *Kitchenham (2004)*, also our study significantly includes research papers that have undergone a rigorous peer-review process, which is a well-established requirement for high quality publications, so the selected paper may be representatives to define the state-of-the-art of microservices granularity definition. For each paper obtained from the query strings, the reason why it was included or excluded from the review was defined. We did not include grey literature.

By using a systematic method already established and widely used in other reviews, the replicability of our study is guaranteed, and the process was rigorously followed to reduce this threat.

### Internal validity

In order to reduce the researcher bias a pre-defined protocol was defined (See Fig. 1). The classification criteria for the selected papers were carefully selected and they were defined in other literature reviews, these literature reviews were explained in the related work section.

We downloaded the selected papers; they were shared with all authors for review. The papers were summarized, we detailed the contribution and made the classification and analysis based on full-text papers. We specified a paper ID, the used technique, the input data, the full paper summary and synthesis, the description of the proposal, the journal or conference where it was published, and observations and comments. We tabulated the papers using the classification criteria explained in 'Discussion', we review each selected paper based on the interpretation of the contribution raised in each one, then we grouped the papers. To reduce the selection bias this process was reviewed for each author independently.

The threats to data synthesis and results were mitigated by having a unified classification and description scheme and following a standard protocol where a systematic process was done and externally evaluated. The data extraction process was aligned with our research questions, also we applied the guidelines of a classic systematic literature review, following a research protocol, thus making our research easy to check and replicate.

## CONCLUSIONS

This systematic literature review identified the main contributions and research gaps regarding the dimensioning and definition of the granularity of microservices comprising an application. Methods, methodologies, and techniques to determine the granularity of microservices were identified.

Microservice granularity research is at a Wild West stage: no standard definition exist, development-operation trade-offs are unclear, there is little notion of continuous granularity improvement, and conceptual reuse is scarce (*e.g.*, few methods seem applicable or replicable in projects other than the first to use them). These gaps in granularity research offer clear options for research on continuous improvement of the development and operation of microservice-based systems.

We propose a microservice granularity definition, first by its size or dimensions, meaning the number of operations (services) exposed by the microservice, along with the number of microservices that are part of the whole application and by its complexity and dependencies. The goal is to have low coupling, low complexity, and high cohesion of the microservices. Defining the most optimal granularity for microservices can significantly improve performance, maintainability, scalability, network use and consumption, computational resources, and cost, because microservices mainly are deployed in the cloud.

As future work we will propose "Microservice Backlog" as a model and techniques to define and evaluate the microservice granularity at design time, using metrics to evaluate the granularity. We want to develop a genetic programing technique and a semantic grouping algorithm to group the user stories of the product backlog into candidate microservices, so the architect or development team can evaluate the candidate decomposition of the application.

### Funding

This work was supported by Colombia's Ministry of Science and Technology (Minciencias-Colciencias) through doctoral scholarship "753-Formación de capital humano de alto nivel para el departamento Norte de Santander"; by the Francisco de Paula Santander University (Cúcuta, Colombia) through the doctoral studies commission number 14 of 2016; by the Universidad del Valle (Cali, Colombia); and by ANID (Chile) through PIA/APOYO AFB180002. There was no additional external funding received for this study. The funders had no role in study design, data collection and analysis, decision to publish, or preparation of the manuscript.

### Grant Disclosures

The following grant information was disclosed by the authors:
Colombia's Ministry of Science and Technology (Minciencias-Colciencias) through doctoral scholarship "753-Formación de capital humano de alto nivel para el departamento Norte de Santander".

Francisco de Paula Santander University (Cúcuta, Colombia) through the doctoral studies commission number 14 of 2016.
Universidad del Valle (Cali, Colombia); and by ANID (Chile) through PIA/APOYO AFB180002.

## Competing Interests
The authors declare that they have no competing interests.

## Author Contributions
- Fredy H. Vera-Rivera conceived and designed the experiments, performed the experiments, analyzed the data, prepared figures and/or tables, and approved the final draft.
- Carlos Gaona conceived and designed the experiments, performed the experiments, analyzed the data, authored or reviewed drafts of the paper, and approved the final draft.
- Hernán Astudillo conceived and designed the experiments, performed the experiments, analyzed the data, authored or reviewed drafts of the paper, and approved the final draft.

## Data Availability
The data of the selected papers and their classification are available in the Supplemental File.

## Supplemental Information
Supplemental information for this article can be found online at http://dx.doi.org/10.7717/peerj-cs.695#supplemental-information.

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
