# Peer review of "Defining and measuring microservice granularity—a literature overview"

_PeerJ Computer Science, doi:10.7717/peerj-cs.695_

## Round 0.1 · original submission · Major Revisions

As you can see from the detailed review reports, the reviewers see value in this study, but have quite a few concern related to the presentation of the paper. In my opinion, the following concerns are particularly salient and should be prioritized in your revision:

- Improving the overall structure (mostly R1, but all reviewers suggest some structural improvements)

- Integrating and condensing (shortening) the initial sections of the paper (R1 and R3)

- Rewriting the threats to validity section (R3)

- Improving the description of the research method (mostly R3, but all reviewers had some methodological concerns)

- Substantially improving the related work section (R3). Particularly note that for a literature review, it is not uncommon to have a very short related work section, since the topically "related" research is discussed extensively in the main part of the paper.

Additionally, please carefully study the comments by all three reviewers and address them as you see fit.

When re-submitting, please submit a version with changes clearly demarked along with your new manuscript. Further, please append a rebuttal letter where you discuss what comments you have addressed and how, and explain which comments you disagree with (and why).

Reviewer 1 ·

Basic reporting

Summary of the paper:
The study reports the state-of-the-art in research about defining and measuring granularity in microservices. To do so, the authors conduct a systematic literature review and try to 1) develop a synthesis of proposed approaches to define microservices granularity from literature, 2) Identify the metrics that are used to evaluate microservice granularity and 3) identify the quality attributes that research about microservices granularity addresses. The review ultimately includes the detailed analysis of 29 papers that investigate the development of microservices systems and how their microservices are created or extracted. The analysis of these papers first of all results to a categorization of the selected studies based on where they were published (journal or conference), phase that papers cover from the microservices development, what type of microservices development is (migration or otherwise), what was the research strategy, what type of contribution the paper resulted to and what research methodologies were used. Then, it provides an overview of the current contributions and gaps of the research landscape about defining granularity of microservices. This is done by organizing the papers based on the microservices development technique, type of contribution / outcome of papers and level of automatization. Further, the study identifies 67 metrics that are used in different studies to evaluate granularity. These metrics are themed based on coupling, cohesion, complexity, performance, other quality attributes, computational resources, team, source code and granularity. Finally, the authors give an overview of quality attributes in defining microservices granularity. These are regarding, runtime, software as an artifact and quality attributes and AI.

An important improvement potential is the amount of text and reading time that is needed until reaching the objectives and Research Questions of the study. It is not easy to make the links between the motivation points, challenges, research gaps, objectives of the study and research questions. Therefore, i suggest for the research questions to be moved closer to the text that motivates the study and states the challenges and research gaps addressed.

Another aspect that this paper has large potential for improvement is structure. The division of sections can be organized differently to improve readability and understandability of the content. Specifically, section 2 and section 3 seem to have a very similar purpose of providing the required Background in the available literature and on drawing the theoretical landscape of the topic. Hence, maybe they could be either merged or explicitly describing their respective value to justify and motivate their separation. In addition, section 5 even though it is organized nicely based on the objectives of the study, the separation of descriptive results and reflections or discussion on concepts would make it easier to read through and understand. For example,in the "Classification of the selected papers", information / data gathered are described about the findings from reviewing the literature in the first and last 2 paragraphs, split by a paragraph that discusses and reflects upon migrations, agile practices and DevOps. I suggest that each piece of result is either presented and then discussed or reflected upon consistently, or all results are stated and then all reflections and discussions are made separately. The in-between format that is currently present is not optimal.

Finally, the language used is generally good and unambiguous with some minor errors, but generally the paper is well written. Some more minor points for improvement are:
- The past-sentence is used in the paper sometimes inconsistently which makes it hard to comprehend. The paper can be written entirely in present-sentence
- In section 2 and 3 the literature is reviewed in a very linear fashion, with small summaries of each study next to each other, supported by quotes. This is good but can be improved my synthesizing further the concepts from the different studies and making a more detailed review of the contents.
- In-text reference formatting is often incorrect. For example, in line 115: instead of "(Hassan, Bahsoon & Kazman, 2020) stated that a granularity...." should be
"Hassan, Bahsoon & Kazman (2020) stated that a granularity....". I suggest you review the referencing formatting more thoroughly in the entire text.
- Titles in section 5 are very long and they could be shortened.

Experimental design

Overall, the content of the paper is relatively good, with a satisfying number of studies involved in the review. This leads to a relatively comprehensive overview of the current state-of-the-art. However, there are some aspects that can be improved further. First of all, the topics that are researched, could be specified and introduced earlier in the paper. So, it should be clear what the objectives are, what are the research questions and then describe how this study is designed to address these. The objectives and research questions should be the starting point of justifying the methodological choices made. In addition, this would make clear why is important to answer the research questions asked in this study and thus, to motivate further the study choices.

Also, the queries that guide data extraction are nicely put and clear. However, a description presenting how they came into surface would provide clarity on why they are the right search terms. Some of them seem to be different and it is not clear what is the impact of that. For example, the term "decomposition" seem to me not equivalent to the rest of the search terms used and hence, how this affects the queries and their accuracy? Furthermore, the selection criteria are clear but their reasoning is not explicitly justified. Some elaboration to describe why are those criteria selected, what other criteria were considered, why other criteria were not selected and what are other potentially valuable criteria would make the authors' rationale more clear. Also, the selection of the papers is stated to be for studies that implicitly or explicitly discuss about granularity levels. This should be further discussed, both methodologically, but also in threats to validity. Finally, more details about the analysis conducted on the content of the chosen papers could be described further, to make clear how the knowledge was utilized.

Validity of the findings

On the content of the research questions, i suggest to further motivate why we need to answer these questions, where are they going to specifically contribute and most importantly if this is the kind of study we currently need in the field. Having that said, one can argue that most of the studies included in the review do not explicitly investigate granularity (granularity is not their central point that they study). They investigate other matters of microservices, like migrations or decompositions and discussion eventually brings some points about granularity. Hence i am reluctant to fully accept that a review on granularity specifically is what we need. I cannot see that enough papers specifically investigate granularity in depth and therefore, it is not clear to me how we can extract valid definitions and metrics about granularity of microservices from the studies included. Most of the studies discuss about granularity and state it as a challenge but not many investigate in depth how granularity is decided. Hence, the gap identified in this paper is valid but the reasons about why this type of research is accurately covering this gap is not clear.

Another very important issue with the presented study is on the analysis of the observations from the reviewed literature. There is room for further analysis to present deeper insights into the concepts. While the results are doing a good job in describing the definitions, metrics and quality attributes for granularity, it would be very insightful to discuss more relations and also differences between different studies. For example, the differences between different categories of papers on the metrics that they use to evaluate granularity. Furthermore, even though the metrics are organized in metric types, they could be analyzed more consistently and systematically, to make clear how impactful they seem to be (e.g. some metrics can be seen in many papers or some metric types appear more frequently). Deeper analysis of the results can be proven to be very insightful and also help making the current results less overwhelming and more organized. This in combination with the need to better motivate the research questions and adjust the research questions to be more accurate in addressing objectives and research gaps, leads me to my final verdict below.

Reviewer 2 ·

Basic reporting

Abstract.
Some results are not related to the research questions of the survey. For example,
“The most used approaches were machine learning, semantic similarity, genetic programming, and domain engineering. Most papers were concerned with migration from monoliths to microservices; and a few addressed greenfield development, but none address improvement of granularity in existing microservice-based systems.”
Also, the quantitative results (as applicable) should be reported (e.g., number of metrics found)
Some conclusions are also not related to results and research questions.
Introduction
It is not clear why the “granularity” important topic and why research questions are important and relevant. Also, the summary of the findings should be included
Some statements are wrong/unjustifiable. SOA was always a style for loosely-coupled distributed systems
“SOA and web services are more related to monolithic applications that ….”
Previous work => Related work
Summary of some related works needs to be simplified (without simply coping the research questions used by those studies).
“(Ghofrani & Lübke, 2018) focused on solving the following questions …….”
The relevance/importance/relationship of “”granularity”, “metrics”, and “quality attributes” need to be highlighted. This would lead to research gaps and questions.

Experimental design

Survey methodology
In general. The author follows the standard literature review methodology.
Why does RQ1 include “microservice granularity and determine the microservices’ size?” Is “size” not part of “granularity”?
Not sure how the keyword “dimensioning” is relevant
The inclusion/exclusion criteria should be listed clearly.
Results and discussions
The results, in particular metrics (RQ2), seems comprehensive.
There are some mismatches between RQ1 and the results in Section 5.2. As per RQ1, 5.2 should report “approaches to defining microservice granularity and size”.
As per RQ3, the reader will expect to see the relationship between the microservice granularity and quality attributes in Section 5.4. But, this perspective is not clear. In generals, the results should be related to RQs
There is no section on discussion of the results and their implications. A discussion section should be included.
Threats to validity can be more elaborated and made comprehensive. The authors are advised to study Threats to validity reported in the good existing survey papers.

Validity of the findings

A comprehensive catalog of microservice metrics are important. However, the application of these metrics should be highlighted.

Reviewer 3 ·

Basic reporting

Monolithic software systems are constructed as a single logical unit that encapsulates business functionalities, which share the same computational resources (e.g., memory space, CPU, and database). This architectural style has been used widely by companies since it provides a uniform and standardized codebase written in the same programming language that facilitates software engineers to
address a large number of cross-cutting concerns such as logging, security features, and exception handling. However, monolithic applications increasingly suffer from maintainability, testability, and scalability issues.

The microservices architecture can help to alleviate those problems as each service is supposed to exhibit well-defined functionality, high modularity, and, most importantly, each service should be able to evolve independently. The microservice architecture is an architectural style for building a single application as a collection of fine-grained and autonomous services, each one running in its process and communicating through lightweight mechanisms. In this realm, finding the most appropriate granularity for microservices is still a challenge.

In order to identify the main techniques and methods in the literature that define microservice granularity or that use it in the process of designing microservice-based systems, either from scratch or migrated from monoliths, the paper conducts a systematic literature review (SLR) that aims to answer three research questions: RQ1: Which approaches have been proposed to define microservice granularity and determine the microservices’ size? RQ2: Which metrics were used to evaluate microservice granularity? RQ3: Which quality attributes were addressed when researching microservice granularity? The paper analysed 29 studies published in conferences and journals.

The paper addresses an important topic for both academia and industry and that is in the scope of the journal. It is in general well written, although there are many text issues to be corrected or improved (marked as orange in the attached PDF file) or removed (marked as red in the attached PDF file). Figures and tables are good, but I wonder why they appear only after the references, since this reduced the paper readability.
The references are good. However, in the text, when citing the authors of a paper, you have to use direct citation. For example: “(Zimmermann, 2017) extracted the principles...” (line 155) should be “Zimmermann (2017) extracted the principles…”. Another example: “(Hassan, Bahsoon & Kazman, 2020) stated that a granularity…” turns into “Hassanm, Bashsoon and Kazman (2020) states that a granularity…”. This should be fixed throughout the whole text!

Experimental design

Following I point out my main concerns at each paper section.

1. Introduction and Microservices granularity

I think those two sections should be merged in only one, the Introduction. The whole text of Section 2 could be placed at line 80. This will make clear the definitions of (microservice) granularity just after the authors affirm that that is still a challenge (lines 75 and 76). In addition, it will be easier to see the arguments for why addressing such research topic, i.e., the work’s motivation (currently lines 125 – 142). With this, the content of lines 81-86 should be adjusted to reference the body of text that will be introduced just before it. I also suggest stating the research questions after mentioning them at the Introductions (line 91).

2. Previous work

I didn’t understand what the criteria were to include the papers in that section. As you are presenting an SLR, I was expecting that related work were papers that conducted an SLR on other topics of microservices, showing that none of them has approached the subject of granularity, or even discussing grey literature about microservice granularity. If the authors want to bring a broader discussion on different topics (application modelling and architecture; design and development patterns; industrial adoption; state of practice; grey literature review; and analysis and interviews with industry leaders, software architects, and application developers of microservices-based applications – lines 146 to 149), then more studies on those topics should be described in this section. For instance, I missed work about migration to microservices, since this is one of the main situations in which a more precise/appropriate microservice granularity impacts the system evolution process.

3. Survey Methodology

Since the paper describes an SLR, I think the methodology could be more detailed. The authors followed a very classic schema proposed by Kitchenhan (2004). However, I also suggest the authors reading the following papers:
- Kai Petersen, Sairam Vakkalanka, and Ludwik Kuzniarz. 2015. Guidelines for conducting systematic mapping studies in software engineering: An update. Information and Software Technology 64 (2015). http://www.sciencedirect.com/science/article/pii/S0950584915000646.
- He Zhang, Muhammad Ali Babar, and Paolo Tell. 2011. Identifying relevant studies in software engineering. Information and Software Technology 53, 6 (2011). http://www.sciencedirect.com/science/article/pii/S0950584910002260

I think this section could be divided into (i) Planning the Mapping, where the authors could describe the Research Questions, Search Strategy, Study Selection and Data extraction; and (ii) Conducting the mapping, in which the process defined in the planning phase is executed and explained step-by-step.

About the methodology, I still have some questions/concerns:

- Why did not the authors search other bases, such as SpringerLink and ScienceDirect?
- Was the snowball technique used? If not, why?
- I would expect more detailed exclusion criteria, such as: (i) studies published only as a (short) abstract; (ii) studies not written in English; (iii) studies to which it was not possible to have access; (iv) duplicate studies;
- How did the authors resolve possible conflicts during the paper reading step?

4. Results

- Lines 365-366: “…and the remaining almost-half (14 of 29) used real-life case studies, thus achieving better validation”. What do you mean by better validation? Is that your feeling or something proved?
- About the size, dimension or granularity of microservices, the authors provide 6 possible metrics (lines 387-392). However, I missed metrics regarding the number of methods or exposed interfaces. Wouldn’t that be included?
- Lines 414-415: “Most papers proposed manual procedures to identify the microservices (15 papers);”. Shouldn’t it be “identify microservice size”?
- Lines 461-462: “Several interesting works have been proposed, but there are still few specific, actionable proposals”. What do you mean by “still few specific, actionable proposals”?
- I liked the discussion of RQ1. However, how is it related to the analysed studies (P1 to P29)? For instance, how or which of those work address microservice decomposition, patterns, migration, and microservice-based application developed from scratch? Even though Table 5 summarizes this data, it has to be explained in the text. The authors did that when answering RQ3.
- Regarding answering RQ2, I would like to know which metrics are new and which ones are reused or adapted from general-purpose or other architectures, like SOA. Furthermore, it would also be very interesting if the authors could explain in which situation (e.g. migration, decomposition, development from scratch…) each metric has been used.
- Some metrics are very general, such as logical coupling and semantical coupling. I wonder how those metrics have been applied to the microservice world, as the authors did in the dependency weight metrics. For instance, I am very curious to see how the metrics Function Point (line 689) has been used to estimate microservice granularity.
- Lines 478-481: The authors said that classified the metrics into four groups, but it seems that there have been found metrics for only two groups. In that case, wouldn’t it be better to propose only two groups rather than four?
- Lines 484 – 485: The authors said that there have been found 14 coupling metrics and 13 cohesion metrics, but the text present 17 and eight metrics of each type, respectively. Please review the numbers for all types of metrics.
- Please remove the period at the end of the names of subsections 5.2.1 to 5.2.8
- Line 513: “I change a software component, I must…”. Please use the passive voice.
- Why is “Contributor count and contributor coupling” (line 539) a coupling metrics rather than a team metrics?
- Lines 553-557: Repeated Metrics.
- Do the metrics Number of singleton clusters (line 702) and maximum cluster size (line 707) refer to containers rather than clusters when considering microservices?
- Lines 711-713: Shouldn’t that paragraph be in Section 5.2.2?
- Can the authors explain the difference between the metrics maximum request time (line 732) and maximum response time (line 735)?
- Can the authors explain how the metrics cost of quality assurance (line 749) is related to microservice granularity?
- Lines 762-763: “(Ahmadvand & Ibrahim, 2016) stated that the level of scalability required for each functional requirement or service.” -> Missing something in the sentence to make sense.
- Wouldn’t “Internal co-change frequency” (line 830) and “External co-change frequency” (line 835) be coupling metrics?
- Lines 860-861: “Being more granular implies that microservice has no dependencies and can function only”. I disagree with that. One microservice can be very big in terms of LOC, providing many services related to the domain object, but has no dependencies. Therefore, it would not be granular.
- Where is the weight in the metrics “Weighted service interface count” (line 864)? What is the difference from these metrics to “operation number”, since both of them count the number of exposed public methods?
- Wouldn’t “Number of microservices” (line 875) be a system metrics rather than a microservice metrics?
- What is the difference between the metrics “Code size in lines of code” (line 815), classified as a source code metrics, and “Lines of code” (line 878), classified as granularity metrics?
- What is a nanoentity (lines 881 and 882)?
- Line 889: “Some have already been defined, mainly for migrations from monoliths to microservices”. Please add references to that sentence.
- Lines 918-919: “Security and fault tolerance are key attributes that microservice-based applications must handle, few works addressed these features”. I wonder how security and fault tolerance can be used to design/estimate microservice granularity.
- Line 1045: What is a DFD PS?

Validity of the findings

5. Threats to Validity

This section has to be completely rewritten, since the text there does not present the threats to the validity of the work and how they have been minimized. Usually threats to validity concern external, internal, construction, and conclusion validity. To a more detailed description about that, please refer to the book: Experimentation in software engineering. Springer Science & Business Media. Claes Wohlin, Per Runeson, Martin Höst, Magnus C Ohlsson, Björn Regnell, and Anders Wesslén.

6. Conclusion

I think that section looks like more a summary of the whole paper rather than a conclusion section. Maybe the authors could create a subsection “Summary”, at the end of the Section “Results and Discussion” and move most of the text from this section to that new one. This way, the readers can find more easily what the conclusions of this work are.
Lines 1114 – 1116: “Finally, the main reason to migrate a monolithic application to microservices is precisely to improve performance and scalability, followed by fault tolerance, maintainability, and modularity”. Where did you get that information from? Could please cite some references?

Finally, although the authors mentioned research gaps, I would like to see possible future work of the presented paper.

Additional comments

The paper is interesting, but it needs some improvements. Please see my suggestions.

Annotated reviews are not available for download in order to protect the identity of reviewers who chose to remain anonymous.

---

## Round 0.2 · Minor Revisions

As you can see, the reviewers and I are largely very happy with the revised version of the manuscript. Reviewer 3 has identified a small number of necessary revisions. I encourage the authors to address these (which should be very quick) and resubmit.

No further round of external review will be necessary for these small changes, so I am confident that we can move towards acceptance very quickly after these changes.

Reviewer 2 ·

Basic reporting

This revision seems to have addressed my review comments for the previous revision adequately.

Experimental design

The study design in this revision is rigorous, and include all key steps in a standard systematic literature review process.

Validity of the findings

In this revision, the research questions, results, and conclusions are properly aligned

Additional comments

This revision seems to have addressed my review comments for the previous revision adequately.

However, as the authors have not used a different text color for changes being done, it is hard to check the exact differences.

Reviewer 3 ·

Basic reporting

The authors considerable improved the paper by answering most of the reviewers' questions and accepting their text suggestions. Particularly, I thank the authors for having addressed my points. Tables and figures have also improved, and new references have been added.

I still found some minor problems in the text, such as:
- Line 216: "this is the first study focus specifically..." => "this is the first study that focuses specifically..."
- Line 221: "...introduced by." => please remove the period.
- Use of references as subjects of sentences, particularly in Section 5.2. For example:
* Line 491: "(Bogner, Wagner & Zimmermann, 2017b) performed..." => "Bogner, Wagner, and Zimmermann (2017b) performed..."
* Line 514: "(Ahmadvand & Ibrahim, 2016) said... "=> "Ahmadvand and Ibrahim (2016) said..."
* Line 649: "According to (Cojocaru, Uta & Oprescu, 2019)," => "According to Cojocaru, Uta, and Oprescu (2019)"
Please read that section through and fix that problem, since it happened so many times.
- Line 720: there is a broken sentence: "It is the"

Experimental design

The study design is now clearer with the improvements in the structure of Section 3. The results in Section 4 were also better described and detailed.

Validity of the findings

Section 5 is a great improvement bringing discussion on the findings and presenting research gaps and trends. In addition, Section 6 is now in accordance with what we expect for a "Threats to validity" section.

Additional comments

I finish by asking the authors to have another round of reading and fixing text problems (some of them are indicated in the first section of my review). Apart from that, I am happy with the new version of the text.

---

## Round 0.3 · accepted · Accept

I have reviewed the (small) changes to the manuscript, and I am happy to accept this submission now. An external review is not necessary anymore at this stage.

Congratulations from my side for an interesting literature review!